# Two duplicated *gsdf* homeologs cooperatively regulate male differentiation by inhibiting *cyp19a1a* transcription in a hexaploid fish

Ming-Tao Wang[1,2], Zhi Li[1,2], Miao Ding[1,2], Tian-Zi Yao[1,2], Sheng Yang[1,2], Xiao-Juan Zhang[1,2], Chun Miao[1,2], Wen-Xuan Du[1,2], Qian Shi[1,2], Shun Li[1,2], Jie Mei[3], Yang Wang [1,2], Zhong-Wei Wang[1,2], Li Zhou[1,2], Xi-Yin Li [1,2]*, Jian-Fang Gui [1,2]*

**1** State Key Laboratory of Freshwater Ecology and Biotechnology, Hubei Hongshan Laboratory, The Innovative Academy of Seed Design, Institute of Hydrobiology, Chinese Academy of Sciences, Wuhan, China, **2** University of Chinese Academy of Sciences, Beijing, China, **3** College of Fisheries, Huazhong Agricultural University, Wuhan, China

* lixiyin@ihb.ac.cn (X-YL); jfgui@ihb.ac.cn (J-FG)

**Data Availability Statement:** The whole genome assembly of *C. gibelio* is deposited at GenBank database under the accession number of under the accession number of PRJNA546443 (BioSample

## Abstract

Although evolutionary fates and expression patterns of duplicated genes have been extensively investigated, how duplicated genes co-regulate a biological process in polyploids remains largely unknown. Here, we identified two *gsdf* (gonadal somatic cell-derived factor) homeologous genes (*gsdf-A* and *gsdf-B*) in hexaploid gibel carp (*Carassius gibelio*), wherein each homeolog contained three highly conserved alleles. Interestingly, *gsdf-A* and *gsdf-B* transcription were mainly activated by *dmrt1-A* (dsx- and mab-3-related transcription factor 1) and *dmrt1-B*, respectively. Loss of either *gsdf-A* or *gsdf-B* alone resulted in partial male-to-female sex reversal and loss of both caused complete sex reversal, which could be rescued by a nonsteroidal aromatase inhibitor. Compensatory expression of *gsdf-A* and *gsdf-B* was observed in *gsdf-B* and *gsdf-A* mutants, respectively. Subsequently, we determined that in tissue culture cells, Gsdf-A and Gsdf-B both interacted with Ncoa5 (nuclear receptor coactivator 5) and blocked Ncoa5 interaction with Rora (retinoic acid-related orphan receptor-alpha) to repress Rora/Ncoa5-induced activation of *cyp19a1a* (cytochrome P450, family 19, subfamily A, polypeptide 1a). These findings illustrate that Gsdf-A and Gsdf-B can regulate male differentiation by inhibiting *cyp19a1a* transcription in hexaploid gibel carp and also reveal that Gsdf-A and Gsdf-B can interact with Ncoa5 to suppress *cyp19a1a* transcription *in vitro*. This study provides a typical case of cooperative mechanism of duplicated genes in polyploids and also sheds light on the conserved evolution of sex differentiation.

## Author summary

Polyploidy generates extra chromosome sets and duplicated genes. However, how the duplicated genes co-regulate a biological process in polyploids remains largely unknown. Here, we reveal that two *gsdf* (gonadal somatic cell-derived factor) homeologs (*gsdf-A* and *gsdf-B*) cooperatively induce male differentiation by inhibiting *cyp19a1a* (cytochrome

SAMN11978084). And other data is available in the main text or supplementary information.

**Funding:** This work was supported by the National Key Research and Development Project (2018YFD0900204, L.Z.; 2021YFD1200804, X.Y. L.), the National Natural Science Foundation of China (31873036, X.Y.L.), the Key Program of Frontier Sciences of the Chinese Academy of Sciences (QYZDY-SSW-SMC025, J.F.G.), the Strategic Priority Research Program of the Chinese Academy of Sciences (XDA24030104, J.F.G.), China Agriculture Research System of MOF and MARA (CARS-45-07, J.F.G.), the Autonomous Project of the State Key Laboratory of Freshwater Ecology and Biotechnology (2019FBZ04, J.F.G.), and the Youth Innovation Promotion Association CAS (2020334, X.Y.L.). The funders had no role in study design, data collection and analysis, decision to publish, or preparation of the manuscript.

**Competing interests:** The authors have declared that no competing interests exist.

P450, family 19, subfamily A, polypeptide 1a) transcription in hexaploid gibel carp (*Carassius gibelio*). Loss of either *gsdf-A* or *gsdf-B* alone results in partial male-to-female sex reversal and loss of both causes complete sex reversal, which can be rescued by a nonsteroidal aromatase inhibitor. Compensatory expression of *gsdf-A* and *gsdf-B* is observed in *gsdf-B* and *gsdf-A* mutants, respectively. Moreover, we determine that *in vitro*, Gsdf-A and Gsdf-B both interact with Ncoa5 (nuclear receptor coactivator 5) and block Ncoa5 interaction with Rora (retinoic acid-related orphan receptor-alpha) to repress Rora/Ncoa5-induced activation of *cyp19a1a*. These findings reveal the potential molecular mechanisms underlying *gsdf* homeologs-mediated male differentiation in polyploid fish.

## Introduction

Polyploidy or whole-genome duplication (WGD) provides extra substrates for genomic evolution and is thus considered to be an important driving force for genetic diversity, trait innovation, and ecological adaption [1–5]. The majority of plants and vertebrates have evolved from polyploid ancestors [6]. Recent polyploidy is also widespread in fishes and amphibians, though it is apparently less frequent than in plants [5,7]. Polyploidy may inherit additional set/sets of chromosomes from the same species (autopolyploidy) or from interspecific hybridization (allopolyploidy) [6]. Homologous chromosomes (and the genes they carry) resulting from allopolyploidy are commonly referred to as homeologs (also homoeologs) [3]. During the initial polyploidization, the neopolyploids are thought to experience genomic chaos resulting from the emergence of duplicated genomes [8–11]. During the subsequent diploidization processes, duplicated genes generated from polyploidy will be eliminated or will evolve divergent functions, and their evolutionary fates are closely associated with the interplay of structural and functional entanglement [12,13]. Recent studies in goldfish and common carp suggest that the subgenomes resulting from the allotetraploidy have continuously rediploidized in a manner from asymmetrical evolution to diverse stabilization [14–16]. The homeologs from subgenomes A and B are co-expressed in most pathways, and their expression dominance shifts temporally during embryogenesis [14]. However, the molecular mechanisms underlying how the duplicated genes co-regulate a biological process in polyploids remains largely unknown.

Except for the two rounds (1R and 2R) of WGD that the common vertebrate ancestor has undergone [17], a fish-specific WGD (3R) is believed to result in the dramatic radiation of teleosts [18, 19]. Intriguingly, the hexaploid gibel carp (*Carassius gibelio*), a cyprinid fish with a wide distribution across Eurasia [20,21], have undergone two extra rounds of polyploidy [22]. An early allotetraploidy about 10–15 Mya resulted in the formation of tetraploid *C. auratus* (AABB, 4n = 100) [14, 23], and a late extra autotriploidy from an ancestral tetraploid approximately 0.5 Mya led to the occurrence of hexaploid gibel carp (*C. gibelio*) (AAABBB, 6n≈150) [22,24]. The hexaploid gibel carp is actually an amphitriploid with two triploid sets of chromosomes derived from both ancestors [22,25,26]. Thus, in hexaploid gibel carp, most genes usually have a total of two homeologs, one from subgenome A and the other from subgenome B, and each homeolog commonly has three alleles, thereby providing an ideal model to investigate cooperative mechanisms of duplicated genes.

The amphitriploid gibel carp has overcome the meiotic obstacle caused by three homologous chromosomes via unisexual gynogenesis, in which the eggs are activated by the sperm of sympatric sexual species (kleptospermy) to initiate embryogenesis using only maternal genetic information [27,28]. In contrast to other unisexual vertebrates, variable male proportions ranging from 1.2% to 26.5% have been discovered in wild populations [29,30]. In our previous

studies, we identified a genetic male-specific marker (MSM) in gibel carp [31] and revealed that male-specific supernumerary microchromosomes are closely associated with the occurrence of genotypic males in a dose-dependent relationship [32,33]. When a female gibel carp is mated with a male from other species, a typical gynogenesis is initiated that all the offspring have the same genetic information as the maternal individual. When a female gibel carp is mated with a genotypic amphitriploid male, a variant of gynogenesis is initiated, during which the sperm nuclei are also extruded but some supernumerary microchromosomes of sperm nuclei occasionally leak into the unreduced eggs. This variant of gynogenesis can accumulate microchromosomes, generate males, and create genetic diversity in the offspring [32–35]. However, details concerning the molecular mechanism of male determination and differentiation in this gynogenetic hexaploid fish are limited.

In sharp contrast to the remarkable diversity of sex-determining switches [36–38], the downstream genetic cascades of sex differentiation are relatively conserved [37,39–41]. Members of the transforming growth factor-β (TGF-β) signaling pathway have been identified as being vastly involved in sex determination and differentiation in vertebrates [36,42]. Gonadal somatic cell-derived factor (*gsdf*), a member of the TGF-β superfamily [43], commonly acts as a male gonad factor during sex determination/differentiation in fish species [36,37,44–49], but the molecular details underlying *gsdf*-mediated male determination/differentiation remain elusive. In this study, we choose the hexaploid gibel carp and the *gsdf* gene, as a unique system to analyze the cooperative mechanisms of duplicated genes on polyploid sex differentiation. We identified two divergent *gsdf* homeologous genes and revealed that the *gsdf* homeologs cooperatively regulated male differentiation by inhibiting *cyp19a1a* transcription. And interactive mechanism analyses illustrated that Gsdf-A and Gsdf-B could interact with Ncoa5 to suppress *cyp19a1a* transcription *in vitro*.

## Results

### Characterization of *gsdf* homeologs and alleles

In the hexaploid gibel carp (*C. gibelio*), we identified two divergent *gsdf* homeologs derived from subgenome A (*gsdf-A*) and subgenome B (*gsdf-B*), which were localized to chromosomes A21 and B21, respectively (Fig 1A). As gibel carp reproduce via unisexual gynogenesis without meiotic recombination [26,27], single nucleotide polymorphisms (SNPs) specific to each allele were stable through generations [25]. According to the SNPs, the sequenced fragments of each homeolog could be clearly divided into three types, indicating that *gsdf-A* and *gsdf-B* had three alleles each (Fig 1B). Subsequently, we cloned the coding sequences of three *gsdf-A* alleles and three *gsdf-B* alleles, and found that the average identity between *gsdf-A* and *gsdf-B* was 85.36 ± 0.34%, while the average identities among three alleles of *gsdf-A* and three alleles of *gsdf-B* were 99.53 ± 0.27% and 99.29 ± 0.36%, respectively (S1A Fig). Interestingly, the deduced amino acid sequences of *gsdf-A* and *gsdf-B* were less conserved (average identity = 77.66% ± 0.37%) than their coding sequences (S1B Fig), as most of the differing nucleotides (68.70 ± 1.33%) caused amino acid changes (S1 Table).

Both *gsdf-A* and *gsdf-B* contain five exons and four introns, which is similar to the *gsdf* genes of *C. auratus*, *Danio rerio*, *Ictalurus punctatus*, *Oryzias latipes*, and *Oreochromis niloticus* (Fig 1C). In addition, most genes in the neighborhood around *gsdf* showed conserved synteny in genomic blocks among these fish species (Fig 1D). Subsequently, we performed sequence alignment of the deduced amino acids and ascertained that the Gsdf-A and Gsdf-B of hexaploid *C. gibelio* had conserved TGF-β domains as in other fish species, particularly the seven or eight cysteines in this domain (S1B Fig). Phylogenetic reconstruction showed that Gsdf-A of *C. gibelio* and *C. auratus* were clustered in one clade, while Gsdf-B of *C. gibelio* and *C. auratus*

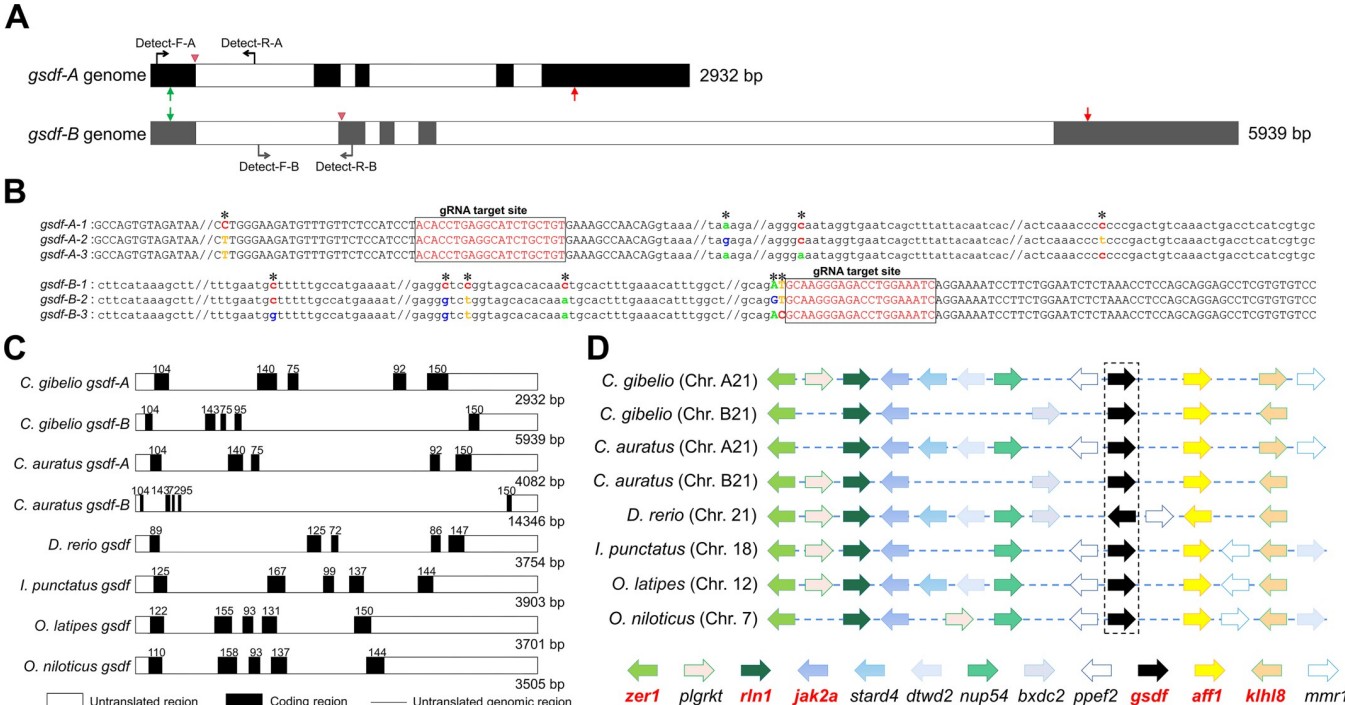

**Fig 1. Characterization of *gsdf-A* and *gsdf-B* in hexaploid gibel carp.** (A) Genomic sequence diagram of two divergent *gsdf* homeologs including *gsdf-A* and *gsdf-B*. The black boxes indicate exons while the white boxes represent introns. Initiation codon and termination codon are marked by green and red arrows, respectively. The primer pairs used to distinguish different alleles and to examine mutant genotypes are indicated by black arrows. Red arrow heads indicate the target sites of guide RNAs (gRNA) used for knockout experiment with CRISPR/Cas9. (B) Fragments of three *gsdf-A* alleles amplified by primer pair of Detect-F-A and Detect-R-A, and fragments of three *gsdf-B* alleles amplified by primer pair of Detect-F-B and Detect-R-B. The asterisks indicate SNPs used for distinguishing different alleles. The red sequences in black box display the target sequences of guide RNAs. (C) Genomic structure of *gsdf* genes among different fish species. Exons and introns are depicted by rectangle boxes and thick lines, respectively. Lengths are exhibited by base pairs (bp). (D) Gene synteny of chromosomal fragments containing *gsdf* genes. Chromosome numbers are displayed at the left side. Conserved gene blocks are represented in matching colors. Transcription orientations are indicated by arrows. The genes marked in red color are conserved genes across all the analyzed species.

were clustered into another clade (S1C Fig), and these patterns were in accordance with the common allopolyploidy origin shared by these two fish species [14,22,26].

## Dynamic transcription of *gsdf-A* and *gsdf-B* are mainly activated by *dmrt1-A* and *dmrt1-B*, respectively

We first examined *gsdf* transcription in eight adult organs via relative real-time quantitative polymerase chain reaction (qPCR) and found out that both *gsdf-A* and *gsdf-B* mRNAs were distributed exclusively in the gonads, with much higher expression in the testis than in the ovary (Fig 2A). Then, we analyzed the dynamic expression profiles of *gsdf-A* and *gsdf-B* in male gonads during developmental stages at 17, 21, 30, 45, 60, 90, 120, 150, 210, 250, 300, and 360 days post hatching (dph). In gibel carp, gonadal morphological differentiation between females and males commonly occurs around 40 dph and gonad mature at about 1 year post hatching [33,50]. The expression of *gsdf-A* steadily increased with a very slow growth rate before 120 dph, displayed a sharp increase from 150 to 250 dph, and then peaked at 250 dph. The expression of *gsdf-B* rapidly increased at 30 dph and peaked at 120 dph. After reaching the peak value, the expression levels of both *gsdf-A* and *gsdf-B* decreased in the mature testis and remained at a certain level (Fig 2B). Subsequently, we produced a polyclonal antibody against Gsdf that could recognize both Gsdf-A and Gsdf-B (S2 Fig), as the deduced amino acid sequences of Gsdf-A and Gsdf-B were relatively conserved (S1B Fig). Immunofluorescence

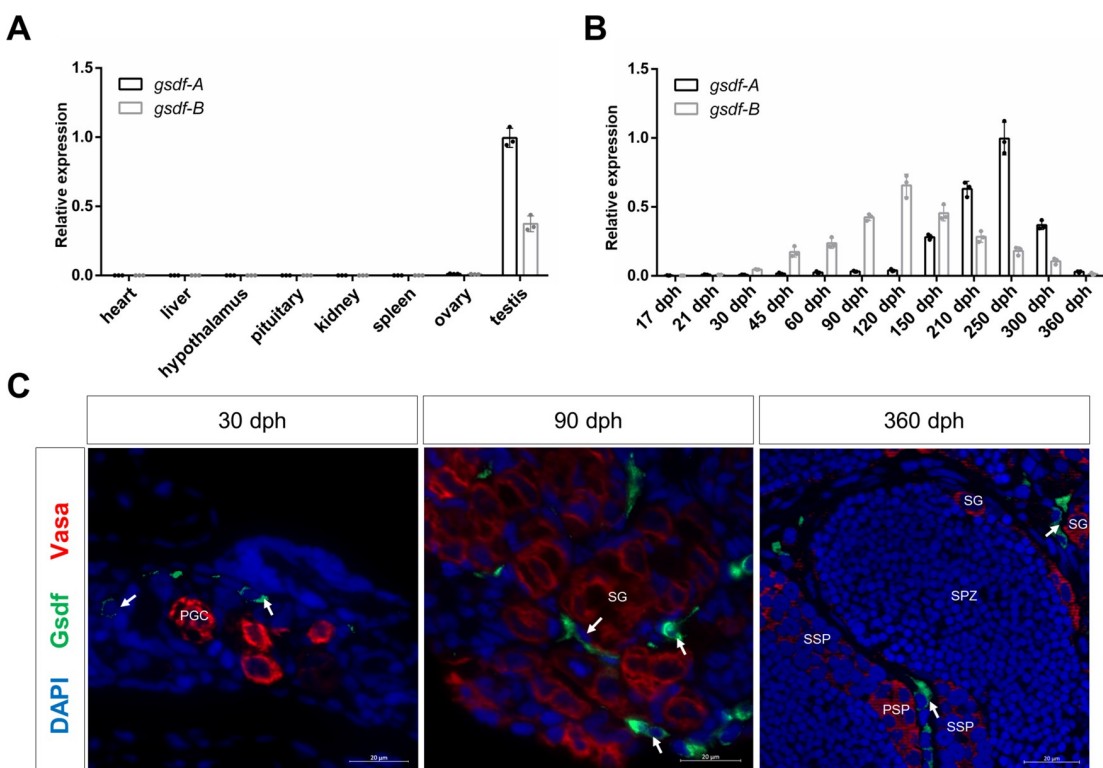

**Fig 2. Expression characterization of *gsdf-A* and *gsdf-B*. (A-B)** Relative real-time quantitative PCR (qPCR) of *gsdf-A*/*gsdf-B* transcripts in adult organs **(A)** and male gonads at different developmental stages **(B)**. **(C)** Immunofluorescence co-localization of Gsdf and Vasa in male gonads at 30, 90, and 360 dph (days post hatching). Green and red fluorescence were immunostained by anti-Gsdf antibody and anti-Vasa antibody, respectively, and blue fluorescence was stained by DAPI. PGC: primordial germ cell; SG: spermatogonium; PSP: primary spermatocyte; SSP: secondary spermatocyte; SPZ: spermatozoa. Arrows indicate somatic cells.

analysis was performed to assess the cellular distribution of Gsdf proteins (Gsdf-A and Gsdf-B) in male gonads at 30, 90, and 360 dph. According to the fluorescence intensity of the anti-Vasa antibody and nuclear morphology, we could easily distinguish germ cells from somatic cells [50,51]. Along with spermatogenesis, the green signal derived from Gsdf was mostly distributed in the cytoplasm of somatic cells surrounding the germ cells (Fig 2C), which was consistent with the main distribution in Sertoli cells of other fishes [46,47,52,53]. Gsdf, as a member of the TGF-β superfamily, is usually considered to be a secreted ligand that will bind to its cell surface receptors [54,55]. However, Gsdf proteins might also accumulate in cytoplasm before being excreted.

By *in silico* analysis, three predicted Dmrt1-binding sites (S3 Fig) and two predicted Sf1 (also named as Nr5a1)-binding sites (S4 Fig) [56] were identified in the upstream sequences of both *gsdf-A* (from -2080 to +50) and *gsdf-B* (from -2150 to +50), which were defined as potential promoter of *gsdf-A* (2130 bp) and *gsdf-B* (2200 bp), respectively (Fig 3A). Subsequently, we cloned these two potential promoters into a pGL3-Basic luciferase reporter vector to analyze Dmrt1's capability for activating *gsdf* promoters. Renilla luciferase plasmid pRL-TK was used as an internal reference. Expression plasmids of Sf1-A/Sf1-B, Dmrt1-A, and Dmrt1-B were constructed, and an empty expression plasmid was used as control. Similar to Nile tilapia [57], gibel carp *gsdf* transcription was also activated by Dmrt1 in a dose-dependent manner in the presence of Sf1 in *Carassius auratus* L. blastulae embryonic (CAB) cells (Fig 3B). Intriguingly, for the *gsdf-A* potential promoter, the transcriptional regulation ability of Dmrt1-A was much

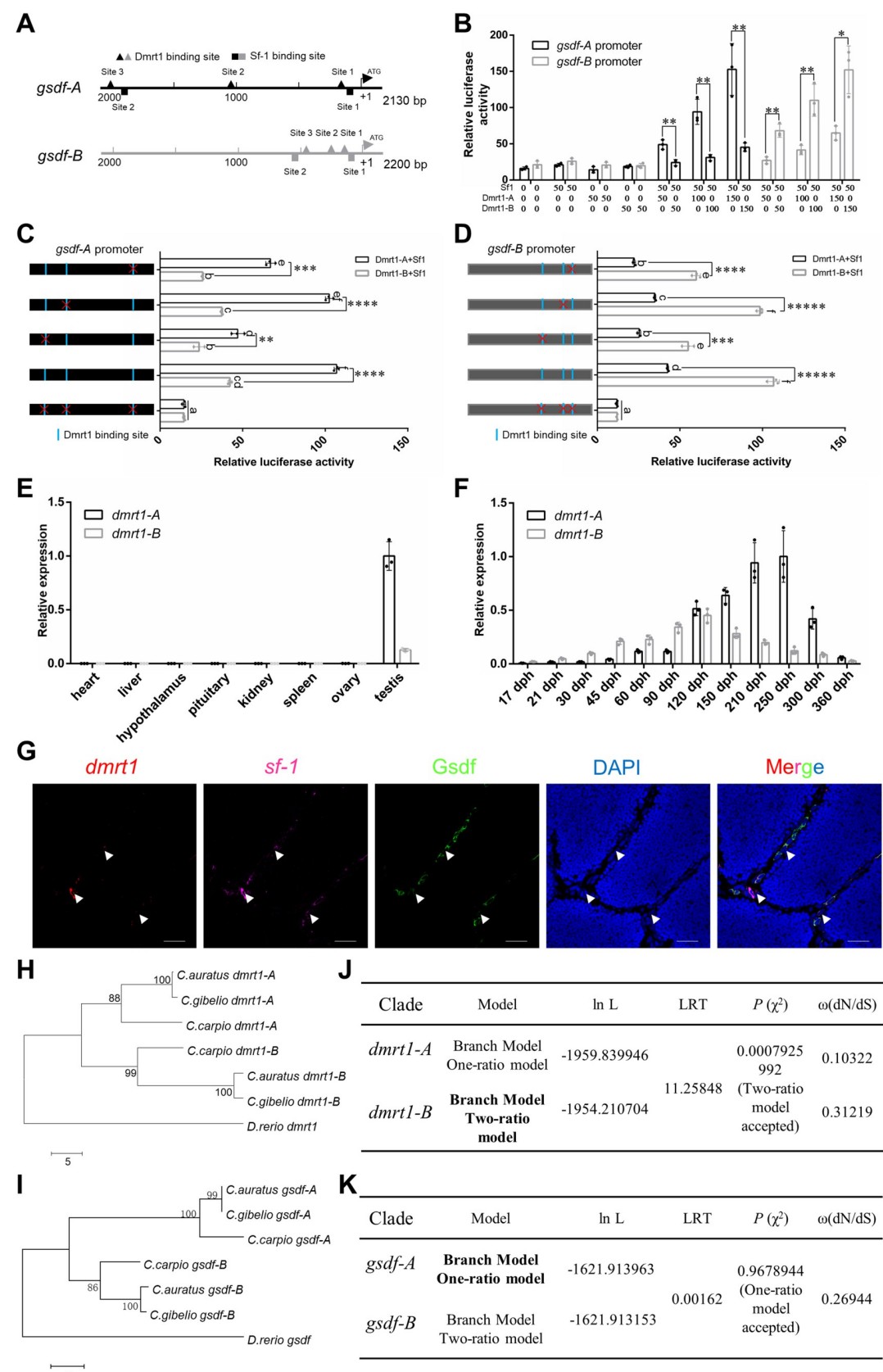

**Fig 3. Dmrt1 regulates *gsdf* transcription in the presence of Sf-1.** (A) Putative Dmrt1 and Sf1 *cis*-regulatory elements in the potential promoters of *gsdf-A* and *gsdf-B*. (B) In the presence of Sf1 (Sf1-A 25 ng + Sf1-B 25 ng), Dmrt1-A/Dmrt1-B overexpression activates the promoter activity of *gsdf-A*/*gsdf-B* in a dosage-dependent manner (50, 100, 150 ng) in CAB cells. For *gsdf-A* potential promoter, the transcriptional regulation ability of Dmrt1-A was significantly stronger than that of Dmrt1-B. For *gsdf-B* potential promoter, the transcriptional regulation ability of Dmrt1-B was significantly stronger than that of Dmrt1-A. The asterisks indicate the significant differences ($*P<0.05$, $**P<0.01$). (C-D) 5′-mutation mapping of Dmrt1-binding region on the *gsdf-A* (C) and *gsdf-B* (D) potential promoter. Blue boxes indicate predicted Dmrt1-binding sites. Red '×' indicates the mutated binding site. The asterisks indicate the significant differences ($*P<0.05$, $**P<0.01$, $***P<0.001$, $****P<0.0001$, $*****P<0.00001$). Different letters represent statistical differences ($P<0.01$). (E-F) qPCR of *dmrt1-A*/*dmrt1-B* transcripts in adult organs (E) and male gonads at different developmental stages (F). The highest expression level in each qPCR analysis was used as control and defined as 1. (G) FISH analysis of the *dmrt1*(Red) and *sf-1* (Pink) mRNA, and immunofluorescence analysis of the Gsdf protein (Green) in mature testis. Arrowhead indicates the somatic cells with expression of *sf-1*, *dmrt1*, and Gsdf. Scale bars: 25 μm. (H-I) Phylogenetic analysis of *dmrt1* (H) and *gsdf* (I) coding sequences from different fish species. Phylogenetic analysis was performed using the Neighbor-Joining method. The percentage of replicate trees in which the associated taxa clustered together in the bootstrap test (2000 replicates) are shown next to the branches. The tree is drawn to scale, with branch lengths in the same units as those of the evolutionary distances used to infer the phylogenetic tree. The evolutionary distances were computed using the Nei-Gojobori method and are in the units of the number of synonymous differences per sequence. The analysis involved 7 nucleotide sequences. All positions containing gaps and missing data were eliminated. (J-K) Information of dN/dS analysis of *dmrt1A/B* (J) or *gsdfA/B* (K). The mode of each gene pairs is selected according to the *P* value ($\chi^2$ test), which is marked in bold. The *dmrt1* or *gsdf* of *D. rerio* were served as references.

stronger than that of Dmrt1-B. For the *gsdf-B* potential promoter, the transcriptional regulation ability of Dmrt1-B was much stronger than that of Dmrt1-A (Fig 3B). Mutation of the Dmrt1-binding site 1 or 3 on *gsdf-A* potential promoter led to a decrease in both Dmrt1-A-induced and Dmrt1-B-induced transcriptional activation of *gsdf-A*. Meanwhile, mutation of the Dmrt1-binding site 1 or 3 on *gsdf-B* potential promoter resulted in a decrease in both Dmrt1-A-induced and Dmrt1-B-induced transcriptional activation of *gsdf-B*. However, mutation of Dmrt1-binding site 2 on *gsdf-A* and *gsdf-B* did not significantly affect Dmrt1-A/Dmrt1-B-induced transcriptional activation (Fig 3C and 3D). Thus, Dmrt1-binding site 1 and 3 of both *gsdf-A* and *gsdf-B* were important for *gsdf* activation.

Subsequently, we found out that the expression patterns of *dmrt1-A* and *dmrt1-B* in both adult organs and in gonads at different developmental stages (Fig 3E and 3F) were closely associated with the expression patterns of *gsdf-A* and *gsdf-B*, respectively (Fig 2A and 2B). Fluorescence *in situ* hybridization (FISH) analysis showed that *dmrt1* and *sf1* mRNA were co-expressed in some Gsdf positive somatic cells in testis (Fig 3G). In addition, we performed dN/dS analyses of *dmrt1* and *gsdf* genes to check whether one or the other homeologs are under selection. Using *D. rerio* as reference, the dN/dS value of *dmrt1-A* was significant lower than that of *dmrt1-B* ($\chi^2$ test *p* value: $7.93 \times 10^{-4}$) under the two-ratio model (Fig 3H and 3J). On the other side, the dN/dS analysis of *gsdf* homeologs met one-ratio model and display the same value (Fig 3I and 3K). These results indicate that the *dmrt1* and *gsdf* homeologs are under asymmetric and symmetric purifying selection, respectively. Thus, the differential expression between *gsdf-A* and *gsdf-B* might be resulted from the divergent evolution of *dmrt1-A* and *dmrt1-B*.

## Deficiency of *gsdf-A* or/and *gsdf-B* leads to partial/complete male-to-female sex reversal

To uncover the function of *gsdf* in male development, we performed loss-of-function analysis using CRISPR/Cas9 in the hexaploid *C. gibelio* with three alleles of *gsdf-A* and three alleles of *gsdf-B*. First, the *gsdf-A* gRNA/Cas9 protein (S5A Fig) or *gsdf-B* gRNA/Cas9 protein (S5B Fig) were injected into the fertilized eggs between a WT female gibel carp without MSM (MSM−) (P: parental generation) and a male common carp to initiate typical gynogenesis [32–35]. All the individuals of the G0 generation were females (MSM−) with *gsdf-A* chimeric mutations

(S5A Fig) or *gsdf-B* chimeric mutations (S5B Fig). Secondly, some of the G0 individuals (MSM −) were mated with a WT male gibel carp with MSM (MSM+) to initiate a variant of gynogenesis, in which some supernumerary microchromosomes of sperm nuclei could occasionally leak into eggs and lead to male occurrence in the offspring [32–35]. The individuals of the G1 generation had various *gsdf-A* genotypes (S5A Fig) or various *gsdf-B* genotypes (S5B Fig). Thirdly, to obtain high proportions of genotypic males in the G2 generation, we chose the sex-reversed phenotypic females (MSM+) from the G1 population as maternal fish (S5 Fig). Two *gsdf-A* mutants with MSM (*gsdf-A*$^{+1/\Delta7/\Delta4}$ + *gsdf-B*$^{+/+/+}$ MSM+, *gsdf-A*$^{\Delta4,\,+9/\Delta4/\Delta7}$ + *gsdf-B*$^{+/+/+}$ MSM+) (S5A Fig) and two *gsdf-B* mutants with MSM (*gsdf-A*$^{+/+/+}$ + *gsdf-B*$^{\Delta2/\Delta1/\Delta2}$ MSM+, *gsdf-A*$^{+/+/+}$ + *gsdf-B*$^{\Delta1/\Delta4/\Delta5,+1}$ MSM+) (S5B Fig) were selected from the G1 population as maternal fish and were separately mated with a WT male gibel carp (MSM+) to construct G2 mutant lines and families. Each symbol separated by a "/" represents one allele of the hexaploid genome. Finally, we established two mutant lines of *gsdf-A* (*gsdf-A*$^{+1/\Delta7/\Delta4}$ + *gsdf-B*$^{+/+/+}$ and *gsdf-A*$^{\Delta4,+9/\Delta4/\Delta7}$ + *gsdf-B*$^{+/+/+}$) (Figs S5A, 4A and 4B) and two mutant lines of *gsdf-B* (*gsdf-A*$^{+/+/+}$ + *gsdf-B*$^{\Delta2/\Delta1/\Delta2}$ and *gsdf-A*$^{+/+/+}$ + *gsdf-B*$^{\Delta1/\Delta4/\Delta5,+1}$) (Figs S5B, 4C and 4D).

In addition, two families of *gsdf-A*/*gsdf-B* double mutants were also established by injecting *gsdf-B* gRNA/Cas9 protein into the fertilized eggs of the *gsdf-A* mutant line (*gsdf-A*$^{\Delta4,+9/\Delta4/\Delta7}$ + *gsdf-B*$^{\text{chimeric mutations}}$) (Figs S5A and 4E) and injecting *gsdf-A* gRNA/Cas9 protein into the fertilized eggs of the *gsdf-B* mutant line (*gsdf-A*$^{\text{chimeric mutations}}$ + *gsdf-B*$^{\Delta2/\Delta1/\Delta2}$) (Figs S5B and 4F). The genotypes of all mutant lines and families were shown in the Fig 4A–4F. As gibel carp reproduce via unisexual gynogenesis without meiotic recombination [26,27], different alleles may have different mutations in one mutant individual [25]. For instance, all the individuals in the *gsdf-A* mutant line (*gsdf-A*$^{+1/\Delta7/\Delta4}$ + *gsdf-B*$^{+/+/+}$ MSM+) had the same genotype where the first allele of *gsdf-A* had a 1-bp insertion, the second allele of *gsdf-A* had a 7-bp deletion, the third allele of *gsdf-A* had a 4-bp deletion, and three alleles of *gsdf-B* were all wild type genotype without mutations (Fig 4A).

Knockout of *gsdf-A* or *gsdf-B* led to partial male-to-female sex reversal (80.6% in line *gsdf-A*$^{+1/\Delta7/\Delta4}$ + *gsdf-B*$^{+/+/+}$, 85.3% in line *gsdf-A*$^{\Delta4,+9/\Delta4/\Delta7}$ + *gsdf-B*$^{+/+/+}$, 83.6% in line *gsdf-A*$^{+/+/+}$ + *gsdf-B*$^{\Delta2/\Delta1/\Delta2}$, and 80.0% in line *gsdf-A*$^{+/+/+}$ + *gsdf-B*$^{\Delta1/\Delta4/\Delta5,+1}$), while disruption of both *gsdf-A* and *gsdf-B* resulted in complete male-to-female sex reversal (Figs S5 and 4G). These results indicate that *gsdf-A* and *gsdf-B* both have male differentiation functions.

## Disruption of *gsdf-A*/*gsdf-B* results in *cyp19a1a* upregulation and aromatase inhibitor (AI) treatment rescues the male-to-female sex reversal

Gonadal morphological differentiation of gibel carp commonly occurs around 40 dph, during which the ovarian cavity is present in female gonads but absent in male gonads [33,50]. We performed analyses of gonadal histology and sex-related gene expression at developmental stages of 25 dph and 55 dph, respectively. At 25 dph, histological examinations in one type of *gsdf-A* mutants (*gsdf-A*$^{\Delta4,+9/\Delta4/\Delta7}$ + *gsdf-B*$^{+/+/+}$ MSM+), one type of *gsdf-B* mutants (*gsdf-A*$^{+/+/+}$ + *gsdf-B*$^{\Delta2/\Delta1/\Delta2}$ MSM+), and two *gsdf-A*/*gsdf-B* double mutants (*gsdf-A*$^{\Delta4,+9/\Delta4/\Delta7}$ + *gsdf-B*$^{\text{chimeric mutations}}$ MSM+ and *gsdf-A*$^{\text{chimeric mutations}}$ + *gsdf-B*$^{\Delta2/\Delta1/\Delta2}$ MSM+) showed that these gonads were similar to WT females (MSM−) and males (MSM+), all of which were at the undifferentiated developmental stage (Fig 5A). Compared with WT male gonads (MSM+), the female differentiation markers *cyp19a1a* and *foxl2b* (forkhead box L2b) were clearly upregulated in the gonads of *gsdf-A* mutants (*gsdf-A*$^{\Delta4,+9/\Delta4/\Delta7}$ + *gsdf-B*$^{+/+/+}$ MSM+), *gsdf-B* mutants (*gsdf-A*$^{+/+/+}$ + *gsdf-B*$^{\Delta2/\Delta1/\Delta2}$ MSM+), and *gsdf-A*/*gsdf-B* double mutants (*gsdf-A*$^{\Delta4,+9/\Delta4/\Delta7}$ + *gsdf-B*$^{\text{chimeric mutations}}$ MSM+ and *gsdf-A*$^{\text{chimeric mutations}}$ + *gsdf-B*$^{\Delta2/\Delta1/\Delta2}$) (Fig 5C). However, the male differentiation markers *dmrt1* and *amh* (anti-Müllerian hormone) were also highly

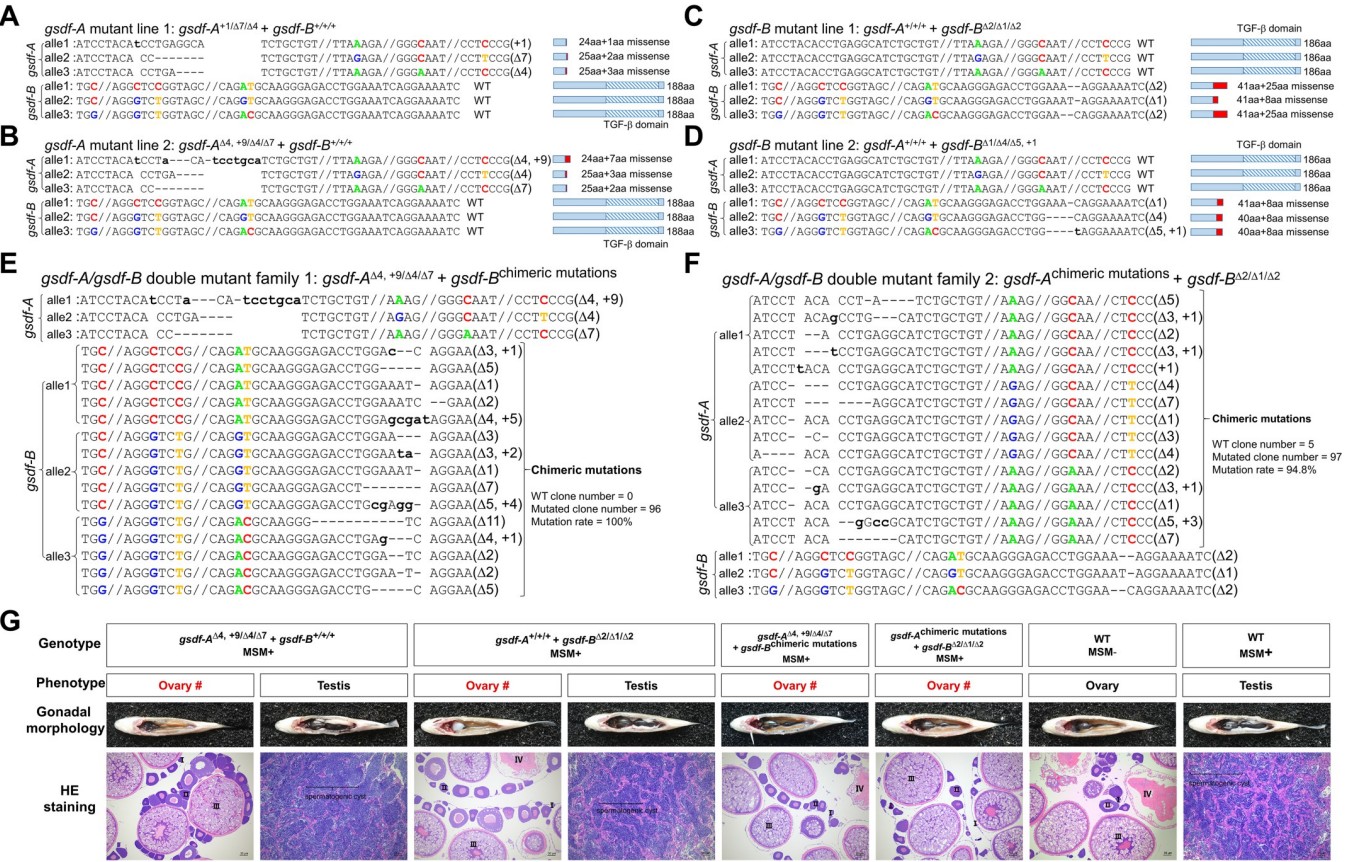

**Fig 4. Genotypes and phenotypes of *gsdf* mutants. (A-F)** Genotypes of *gsdf* mutants: *gsdf-A* mutant line 1 (*gsdf-A*^{+1/Δ7/Δ4} + *gsdf-B*^{+/+/+}) **(A)**, *gsdf-A* mutant line 2 (*gsdf-A*^{Δ4,+9/Δ4/Δ7} + *gsdf-B*^{+/+/+}) **(B)**, *gsdf-B* mutant line 1 (*gsdf-A*^{+/+/+} + *gsdf-B*^{Δ2/Δ1/Δ2}) **(C)**, *gsdf-B* mutant line 2 (*gsdf-A*^{+/+/+} + *gsdf-B*^{Δ1/Δ4/Δ5,+1}) **(D)**, *gsdf-A/gsdf-B* double mutant family 1 (*gsdf-A*^{Δ4,+9/Δ4/Δ7} + *gsdf-B* ^{chimeric mutations}) **(E)**, and *gsdf-A/gsdf-B* double mutant family 2 (*gsdf-A* ^{chimeric mutations} + *gsdf-B*^{Δ2/Δ1/Δ2}) **(F)**. **(G)** Gonadal morphology and histology of adult gonads. #, sex reversal; I, primary oocyte; II, growth stage oocyte; III, vitellogenic oocyte; IV, maturing oocyte. Bars are shown at bottom-right of the images. MSM+, with MSM; MSM−, without MSM; WT, wild type.

expressed in *gsdf-A* or *gsdf-B* single mutants (Fig 5C). Furthermore, the expression levels of primordial germ cell markers *piwil1* (piwi-like RNA-mediated gene silencing 1) and *dnd* (dead end) in these four kinds of *gsdf* mutants (MSM+) were between those in WT males (MSM+) and WT females (MSM−) (Fig 5C). Intriguingly, compensatory high expression of *gsdf-B* was observed in gonads of *gsdf-A* mutants (*gsdf-A*^{Δ4,+9/Δ4/Δ7} + *gsdf-B*^{+/+/+} MSM+), and compensatory high expression of *gsdf-A* was detected in gonads of *gsdf-B* mutants (*gsdf-A*^{+/+/+} + *gsdf-B*^{Δ2/Δ1/Δ2} MSM+) at 25 dph (Fig 5E).

At 55 dph, the gonads of WT females (MSM−) and males (MSM+) differentiated into ovaries and testes, respectively (Fig 5B). Via histological examination, the gonads of *gsdf-A* mutants (*gsdf-A*^{Δ4,+9/Δ4/Δ7} + *gsdf-B*^{+/+/+} MSM+) and *gsdf-B* mutants (*gsdf-A*^{+/+/+} + *gsdf-B*^{Δ2/Δ1/Δ2} MSM+) could be clearly distinguished into ovaries and testes, whereas the gonads of *gsdf-A/gsdf-B* double mutants (*gsdf-A*^{Δ4,+9/Δ4/Δ7} + *gsdf-B* ^{chimeric mutations} MSM+ and *gsdf-A* ^{chimeric mutations} + *gsdf-B*^{Δ2/Δ1/Δ2} MSM+) all developed into ovaries (Fig 5B). Compared with the WT males (MSM+), the expression levels of *cyp19a1a* and *foxl2b* were highly elevated, while the expression levels of *dmrt1* and *amh* were repressed in all the four kinds of *gsdf* mutants (MSM +) with ovaries (Fig 5D). In addition, in the *gsdf-A* mutants (*gsdf-A*^{Δ4,+9/Δ4/Δ7} + *gsdf-B*^{+/+/+} MSM+) and *gsdf-B* mutants (*gsdf-A*^{+/+/+} + *gsdf-B*^{Δ2/Δ1/Δ2} MSM+) with testes, the expression levels of all the examined genes were similar to those in WT males (MSM+) (Fig 5D).

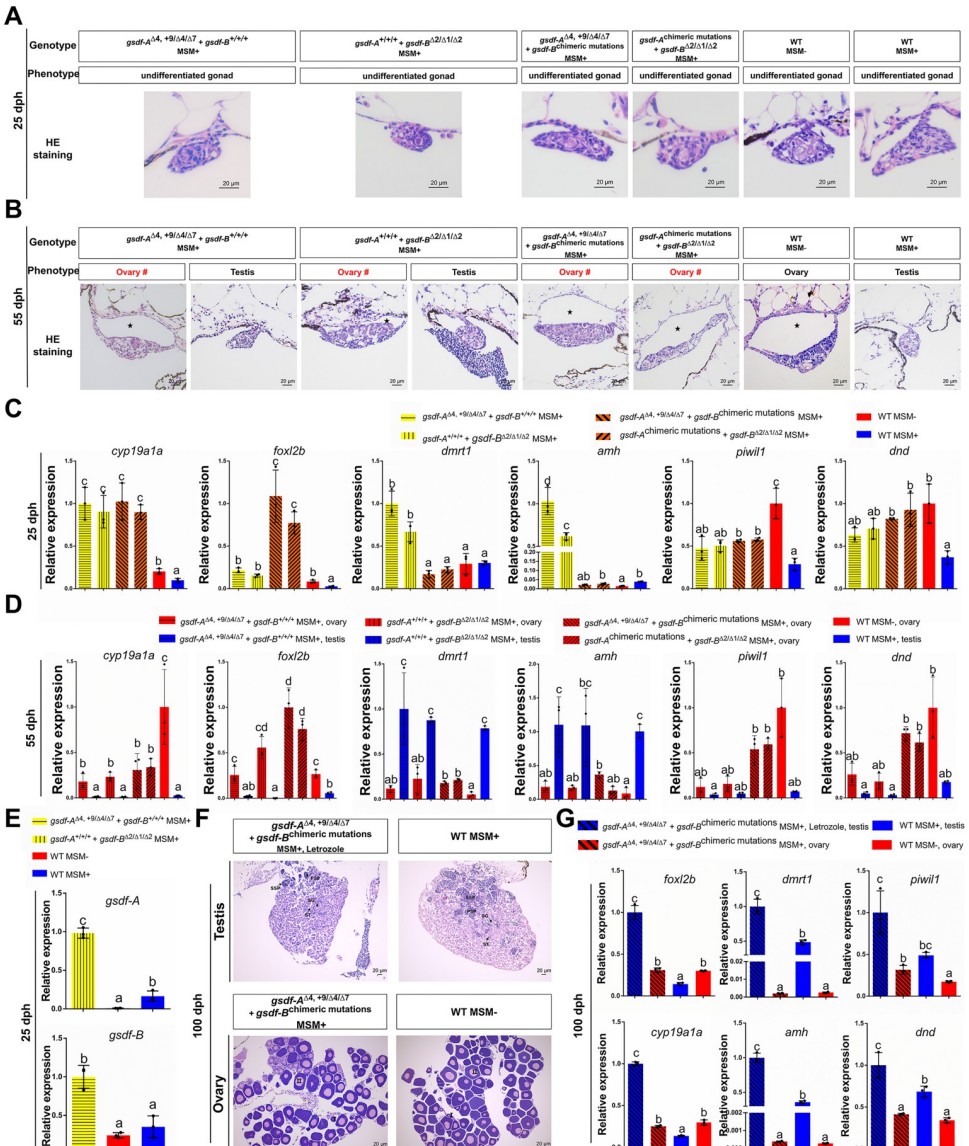

**Fig 5. Aromatase inhibitor treatment rescues the male-to-female sex reversal in *gsdf* mutants. (A-B)** Gonadal histology of different *gsdf* mutants and wild type individuals at 25 **(A)** and 55 dph **(B)**. dph, days post hatching; #, sex reversal; star symbol, ovarian cavity; MSM+, with MSM; MSM−, without MSM; WT, wild type. **(C-D)** Gonadal gene expression of different *gsdf* mutants and wild type individuals at 25 **(C)** and 55 dph **(D)**, including female marker genes *cyp19a1a* and *foxl2b*, male marker genes *dmrt1* and *amh*, and primordial germ cell marker genes *piwil1* and *dnd*. **(E)** *Gsdf-A* compensatory expression in the gonads of *gsdf-B* mutants and *gsdf-B* compensatory expression in the gonads of *gsdf-A* mutants at 25 dph. **(F-G)** Gonadal histology **(F)** and gene expression **(G)** of *gsdf-A/gsdf-B* double mutant individuals (*gsdf-A*$^{\Delta4,+9/\Delta4/\Delta7}$ + *gsdf-B*$^{\text{chimeric mutations}}$, MSM+), *gsdf-A/gsdf-B* double mutant individuals (*gsdf-A*$^{\Delta4,+9/\Delta4/\Delta7}$ + *gsdf-B*$^{\text{chimeric mutations}}$, MSM+) with aromatase inhibitor (Letrozole) treatment, and wild type individuals at 100 dph. Different letters represent statistical differences (*P*<0.05). The highest expression level in each qPCR analysis was used as control and defined as 1.

Although gonads at 25 dph could not be distinguished as ovaries or testes by morphological analysis, the expression levels of marker genes in the *gsdf* mutants, especially the *gsdf-A/gsdf-B* double mutants, could provide clues to sexual fate (Fig 5C and 5D). The gonadal aromatase gene *cyp19a1a* is a conserved factor of ovarian differentiation in fish [58,59] and also has been

demonstrated to play a role in female differentiation in hexaploid *C. gibelio* (S6 Fig). In the undifferentiated gonads of all *gsdf* mutants (MSM+), the ovarian aromatase gene *cyp19a1a* was dramatically upregulated (Fig 5C), indicating that deficiency of *gsdf* might abolish its repression of *cyp19a1a*. Thus, we used nonsteroidal aromatase inhibitor letrozole to analyze whether blockage of Cyp19a1a enzyme activity could rescue the male-to-female sex reversal caused by *gsdf* dysfunction. As expected, administration of letrozole from 15 to 55 dph prevented male-to-female sex reversal in 84.4% of *gsdf-A/gsdf-B* double mutants (MSM+) (Fig 5F). As the period before 15 dph was key developmental stages of sex determination/differentiation, letrozole treatment after 15 dph might be the reason that 15.6% *gsdf-A/gsdf-B* double mutants still developed into phenotypic females. And the gene expression patterns of these letrozole treated testes were similar to that in WT males (MSM+), except for *cyp19a1a* and *foxl2b* (Fig 5G). Upregulation of *cyp19a1a* and its active transcriptional factor *foxl2b* [25,60] in the testis of *gsdf-A/gsdf-B* double mutants (MSM+) subjected to letrozole treatment may have been caused by inhibition of Cyp19a1a enzyme activity. In addition, *cyp19a1a* mRNA was co-expressed with Gsdf in some somatic cells of mature testis (S7 Fig). These results indicate that *gsdf-A* and *gsdf-B* co-inhibit *cyp19a1a* expression, resulting in male differentiation in WT males (MSM+).

## Identification of Gsdf-Ncoa5 interaction via yeast two-hybrid assay and co-immunoprecipitation

As a member of TGF-β superfamily, Gsdf is commonly considered to be a ligand to bind to its cell surface receptors [54]. To identify potential interaction membrane proteins of secreted Gsdf, the coding sequence of Gsdf-A mature peptide was cloned into pBT3-SUC (pBT3-SUC-Gsdf-A mature peptide) as a bait to perform yeast two hybrid assay via DUAL membrane system (Dualsystems Biotech). However, screens did not yield any interactors (S8 Fig). Subsequently, the open reading frame of *gsdf-A* was cloned into pGBKT7 (pGBTKT7-Gsdf-A) as bait to perform yeast two hybrid assay via GAL4 system (Clontech). A total of 239 positive transformants were selected, and their plasmids were isolated for sequencing. These obtained coding sequences belonged to 34 genes according to the genome and transcription data of gibel carp [26,33]. To exclude false positive results, the full-length coding sequences of the 34 genes were cloned into vector pGADT7-AD and co-transformed with *gsdf-A*-pGBTKT7 separately. Finally, a total of 27 proteins were confirmed to be the potential interaction partners of Gsdf-A (Fig 6A and 6B).

We overexpressed these 27 isolated genes in order to identify whether these genes were involved in sex differentiation pathway, by analyzing transcription of sex differentiation genes such as *cyp19a1a*, *foxl2b*, *dmrt1*, and *amh*. In CAB cells, *amh* had no constitutive expression, so we obtained the data of the rest three sex differentiation genes (S9 Fig). Among the 27 proteins, a total of 6 proteins could activate *cyp19a1a* transcription significantly, however, only Ncoa5 activated *cyp19a1a* transcription but did not change expression levels of *foxl2b* and *dmrt1* (S9 Fig). Besides, Ncoa5 was revealed to be involved in the regulating pathway of sex hormones [61,62] and *cyp19a1a* was one of the most important downstream genes of *gsdf* (Fig 5), so we selected Ncoa5 for subsequent analyses. There were two homeologs of *ncoa5* (*ncoa5-A* and *ncoa5-B*) in the gibel carp genome (S10A and S10C Fig) and the protein isolated from yeast two-hybrid assay was Ncoa5-B (Fig 6B). As the protein sequences between Ncoa5-A and Ncoa5-B was highly conserved (identity = 91.53%) (S10C Fig), we only used Ncoa5-B for subsequent *in vitro* experiments.

Subsequently, we found that the presence of Gsdf-A/Gsdf-B could eliminate Ncoa5-induced upregulation of *cyp19a1a* expression (Fig 6C), but could not eliminate Foxl2b/

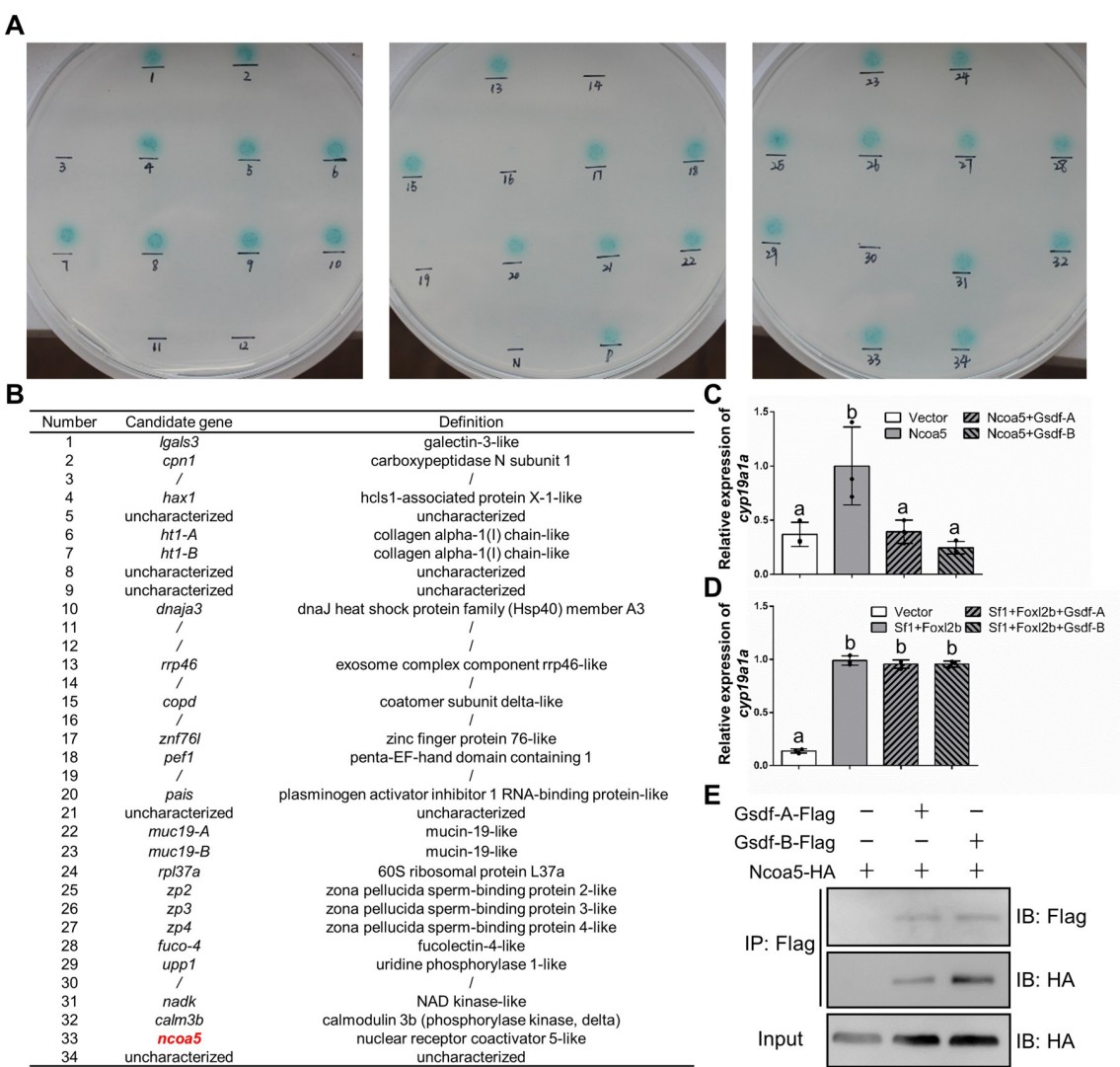

**Fig 6. Identification of Gsdf-Ncoa5 interaction via yeast two-hybrid assay and co-immunoprecipitation. (A)** Confirmation of 34 candidate interacting proteins of Gsdf-A via yeast two-hybrid assay. +: pGBKT7–53 and pGADT7-T co-transformed as positive control; −: pGBKT7-Lam and pGADT7-T co-transformed as negative control. **(B)** List of the 27 potential interaction partners of Gsdf-A. **(C-D)** qPCR analysis of *cyp19a1a* expression in the CAB cells transfected with different plasmids. Different letters represent statistical differences (*P*<0.01). The highest expression level in each qPCR analysis was used as control and defined as 1. **(E)** Co-IP of Gsdf-A-Flag and Gsdf-B-Flag with Ncoa5-HA in CAB cells transfected with the indicated plasmids. Anti-Flag Ab was used for Co-IP. Sf1: Sf1-A and Sf1-B; Foxl2b: Foxl2b-A and Foxl2b-B; Ncoa5: Ncoa5-B.

Sf1-induced upregulation of *cyp19a1a* (Fig 6D). In addition, the interactions of Gsdf-A/Gsdf-B and Ncoa5 were confirmed by co-immunoprecipitation, in which the anti-Flag Ab-immunoprecipitated protein Ncoa5 was recognized by the anti-HA Ab (Fig 6E). These findings indicate that Gsdf-A and Gsdf-B both interact with Ncoa5 to regulate *cyp19a1a* transcription *in vitro*.

## Ncoa5 participates in *cyp19a1a* regulation via interaction with Rora

In humans, Rora is known to interact with Ncoa5 to enhance *cyp19a1a* transcription [61,62]. It would be interesting to know whether gibel carp Rora and Ncoa5 are involved in the expression modulation of *cyp19a1a*. We identified two homeologs of *roraα* (*roraα-A* and *roraα-B*)

and two homeologs of *roraβ* (*roraβ-A* and *roraβ-B*) in the gibel carp genome (S10B and S10D Fig). The sequence identities between Roraα-A/Roraα-B and human RORA was much higher than those between Roraβ-A/Roraβ-B and human RORA (S10D Fig). As the protein sequences between Roraα-A and Roraα-B (identity = 91.88%) was highly conserved (S10D Fig) and Roraα-B had higher identity than Roraα-A compared with human ortholog, we used Roraα-B for subsequent *in vitro* experiments.

In CAB cells, co-immunoprecipitation showed that Rora could interact with Ncoa5 (Fig 7A). In addition, Rora activated *cyp19a1a* transcription in a dose-dependent manner in the presence of Ncoa5 (Fig 7B). Potential Rora-binding sites were predicted in the promoter of *cyp19a1a* (Figs 7C and S11) [63]. Mutation of the potential Rora-binding sites resulted in a decrease in the Rora/Ncoa5-induced transcriptional activation of *cyp19a1a* but did not affect Foxl2b/Sf1-induced upregulation of *cyp19a1a*, indicating that these binding sites were specific to Rora/Ncoa5 (Fig 7D). Subsequently, the Rora-binding site 1 was further confirmed via chromatin immunoprecipitation (ChIP) in CAB cells transfected with Rora-Myc vector. The PCR band containing the binding site 1 of Rora was detected in the chromatin precipitated with the Myc antibody, while no band was observed in the negative control chromatin that was precipitated with nonspecific IgG (Fig 7E). Besides, FISH analysis showed that *rora* and *ncoa5* mRNA were co-expressed in Cyp19a1a positive somatic cells of mature testis (Fig 7F). Thus, these findings indicate that Rora positively regulates transcription of *cyp19a1a* by binding to the promoter in the presence of Ncoa5.

## Gsdf-A and Gsdf-B both inhibit *cyp19a1a* transcription via competitive interaction with Ncoa5

During sex differentiation, expression of *gsdf* was closely negatively associated with expression of *cyp19a1a* in female and male gonads (Fig 8A). The expression of *ncoa5* had no significant difference between females and males in gonads at early developmental stages, while the gonadal expression of *rora* has no difference between females and males before 45 dph but display female-biased expression at 60 dph (Fig 8A). FISH analysis showed that *rora* and *ncoa5* mRNA were co-expressed in some Gsdf positive somatic cells of mature testis (Fig 8B). To elucidate how *gsdf-A* or *gsdf-B* regulates *cyp19a1a* expression, we performed an overexpression analysis, and the overexpression of Gsdf-A or Gsdf-B significantly repressed *cyp19a1a* transcription *in vitro* accompanied by downregulation of *rora* (Fig 8C and 8D). However, the expression of *ncoa5* was not affected by Gsdf-A or Gsdf-B overexpression in tissue culture cells (Fig 8C and 8D), consistent with the *in vivo* result. Meanwhile, the *cyp19a1a* transcription induced by Rora and Ncoa5 was repressed by Gsdf-A and Gsdf-B in a dose-dependent manner (Fig 8E). Furthermore, siRNAs against both *ncoa5-A* and *ncoa5-B* were designed, and the repression of Gsdf-A or Gsdf-B on *cyp19a1a* transcription was abolished by knockdown of both *ncoa5-A* and *ncoa5-B* (Fig 8F). The competitive interaction analysis revealed that Rora recruitment by Ncoa5 decreased upon the participation of Gsdf-A or Gsdf-B but was not affected by the participation of other proteins such as female differentiation factor Foxl2b and immune-related factor Sting (Fig 8G). Thus, these results indicate that the upregulation of Gsdf-A and Gsdf-B increases the interaction of Ncoa5 with both homeologs and reduces the interaction of Ncoa5 with Rora, leading to the downregulation of *cyp19a1a* and subsequent male differentiation.

## Discussion

Polyploidy provides a source of new genes and these duplicated genes will be eliminated/pseudogenized or evolve a sub/neo function during the evolutionary trajectory [2,12,25]. The

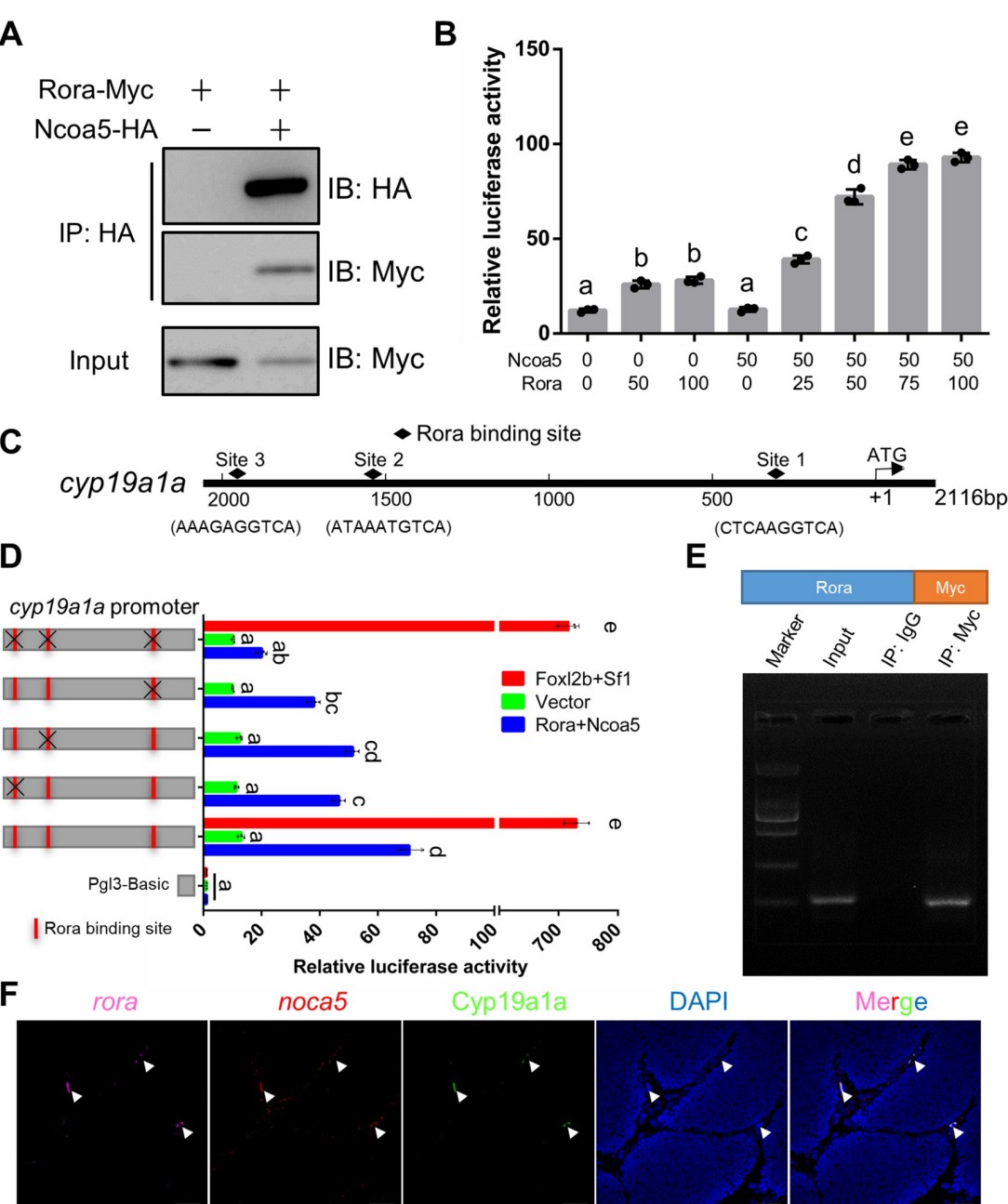

**Fig 7. Rora interacts with Ncoa5 to regulate the transcriptional activity of *cyp19a1a*. (A)** Co-IP of Rora-Myc with Ncoa5-HA in CAB cells transfected with the indicated plasmids. Anti-HA Ab was used for Co-IP. **(B)** In the presence of Ncoa5 (50 ng), Rora overexpression activates *cyp19a1a* promoter in a dosage-dependent manner in EPC cells. Different letters represent statistical differences ($P<0.01$). **(C)** Putative Rora *cis*-regulatory elements in the potential promoter of *cyp19a1a*. **(D)** 5′-mutation mapping of Rora-binding region on the *cyp19a1a* potential promoter. Red boxes indicate predicted Rora binding sites. Black 'X' indicates the mutated binding site. Different letters represent statistical differences ($P<0.01$). **(E)** ChIP-PCR assay in CAB cells transfected with Rora-Myc vector. Specific primers were used to amplify the fragment spanning the Rora binding site on the *cyp19a1a* promoter. The PCR products were detected by 1.5% agarose gel electrophoresis. IgG antibody-based ChIP assay was used as a negative control. **(F)** FISH analysis of the *rora* (Pink) and *ncoa5* (Red) mRNA, and immunofluorescence analysis of the Cyp19a1a protein (Green) in mature testis. Arrowhead indicates the somatic cells with expression of *rora*, *ncoa5*, and Cyp19a1a. Scale bars: 25 μm.

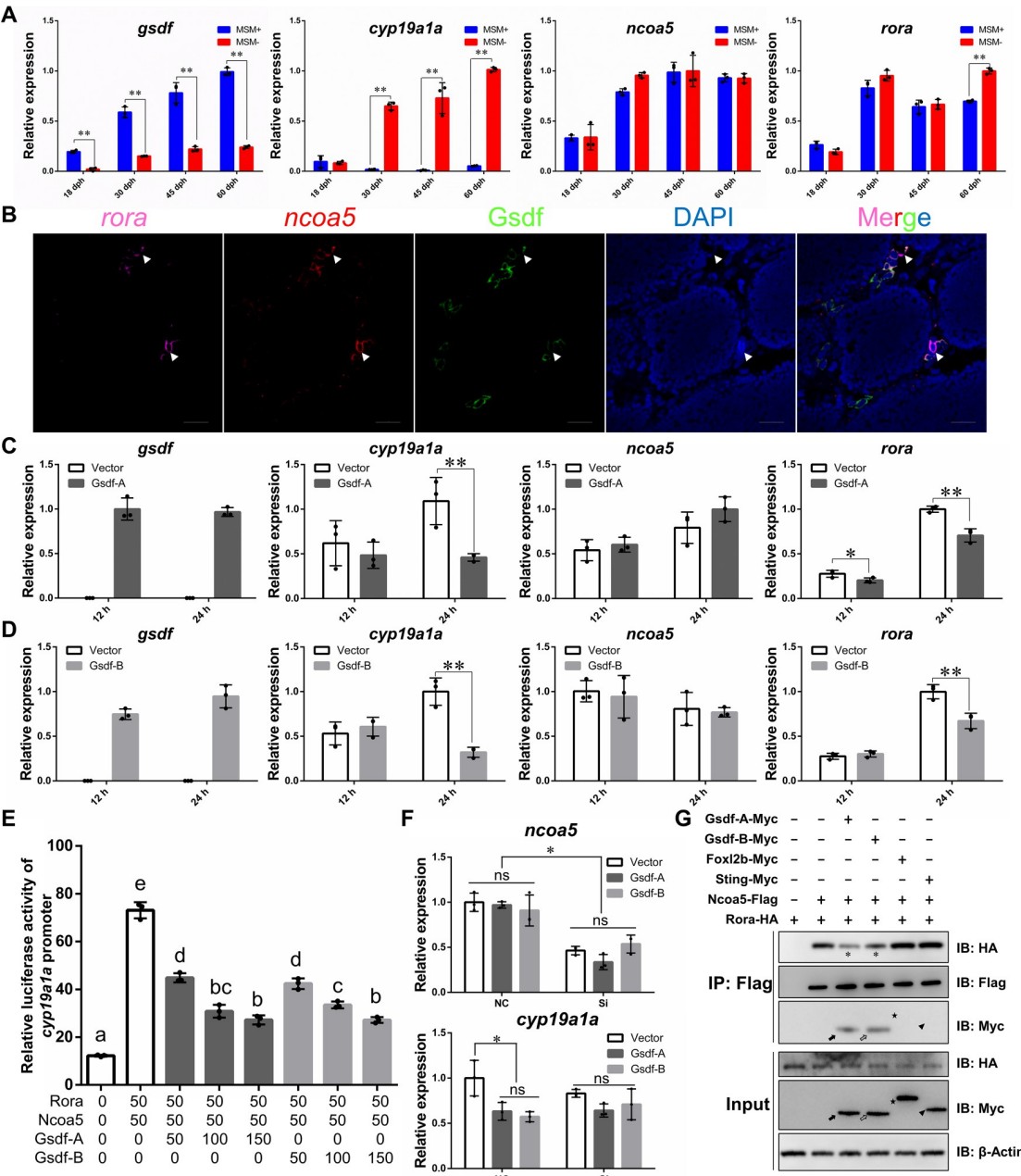

**Fig 8. Gsdf inhibits *cyp19a1a* transcription via competitive interaction with Ncoa5. (A)** Gene expression detected by qPCR in gonads during different developmental stages. MSM+, wild type males with MSM; MSM−, wild type females without MSM; **(B)** FISH analysis of the *rora* (Pink) and *ncoa5* (Red) mRNA, and immunofluorescence analysis of the Gsdf protein (Green) in mature testis. Arrowhead indicates the somatic cells with expression of *rora*, *ncoa5*, and Gsdf. Scale bars: 25 μm. **(C-D)** Gene expression detected by qPCR after overexpression of Gsdf-A **(C)** and Gsdf-B **(D)**. The asterisks indicate the significant differences ($^*P<0.05$, $^{**}P<0.01$). **(E)** In the presence of Ncoa5 and Rora, Gsdf-A/Gsdf-B overexpression inhibits the promoter activity of *cyp19a1a* in a dosage-dependent manner in EPC cells. Different letters represent statistical differences ($P<0.05$). **(F)** Effect of *ncoa5* (*ncoa5-A* and *ncoa5-B*) RNAi on the Gsdf-A/Gsdf-B-induced repression of *cyp19a1a*. qPCR analysis of *ncoa5* and *cyp19a1a* in the EPC cells after transfection with different plasmids and *ncoa5-A/B* siRNA for 24 h ($^*P<0.05$). NC, negative control; Si, *ncoa5-A/B* siRNA. **(G)** Co-IP of Ncoa5-Flag and Rora-HA with Gsdf-A-Myc, Gsdf-B-Myc, Foxl2b-Myc, and Sting-Myc in EPC cells transfected with the indicated plasmids. Anti-Flag Ab was used for Co-IP. Black arrow, gray arrow, star, and arrow head indicate bands of Gsdf-A-Myc, Gsdf-B-Myc, Foxl2b-Myc, and Sting-Myc, respectively. The asterisks indicate the decreased Rora-HA. The highest expression level in each qPCR analysis was used as control and defined as 1.

hexaploid gibel carp (AAABBB) with extra two rounds of polyploidy origins has retained most of the duplicates, where most genes usually have two homeologs, and each homeolog commonly has three alleles [22,25,26,64]. Recently, asymmetrical evolution, homoeologous exchanges, and expression divergence of subgenomes A and B have been observed in allotetraploid goldfish, common carp, and hexaploid gibel carp [14,65]. And we also have presented functional divergence of *foxl2* and *viperin* homeologs in hexaploid gibel carp [25,64]. Here, we found expression divergence between *gsdf-A* and *gsdf-B* (Fig 2A and 2B), but knockout *gsdf-A* or *gsdf-B* displayed similar sex-reversal rates (S5 Fig), indicating the contribution of each homeolog is similar in male differentiation. In addition, disruption of *gsdf-A* or *gsdf-B* triggers highly compensatory expression of *gsdf-B* or *gsdf-A* during the critical period of sex differentiation (Fig 5E), and missing either *gsdf-A* or *gsdf-B* does not give complete sex reversal but missing both does, suggesting that *gsdf-A* and *gsdf-B* cooperatively regulate male differentiation in gibel carp.

An intriguing finding of this study is the revelation of potential molecular rationales underlying male differentiation mediated by *gsdf* homeologs. In male gibel carp with MSM (MSM+), high expression of *gsdf-A* and *gsdf-B* in somatic cells suppresses *cyp19a1a* to induce Sertoli cell development and male differentiation. In female individuals without MSM (MSM−), the low levels of Gsdf-A and Gsdf-B cannot inhibit *cyp19a1a* transcription, leading to estrogen production, granulosa cell development, and female differentiation (Fig 9). Besides, *in vitro* analyses revealed that Gsdf-A and Gsdf-B can interact with Ncoa5 to block Ncoa5 interaction with Rora, inhibiting Rora/Ncoa5-induced activation of *cyp19a1a* (Fig 9). Commonly, as a member of TGF-β superfamily, Gsdf is considered to be a secreted ligand to bind to its cell surface receptors [54]. Here we demonstrated that Gsdf might also have functions in cells. However, we still do not known whether other factors are involved in the interaction between Gsdf and Ncoa5.

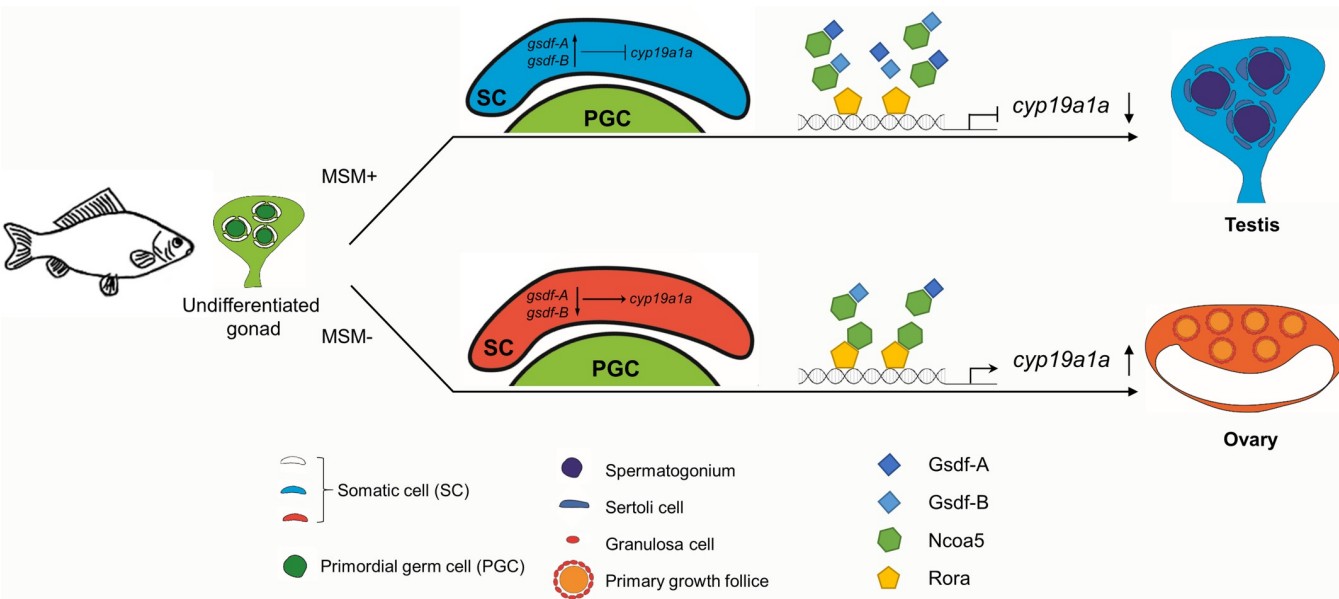

**Fig 9. Hypothetical molecular mechanism underlying *gsdf-A*/*gsdf-B*-mediated male differentiation in gibel carp.** In MSM+ individuals, highly expressed Gsdf-A and Gsdf-B can inhibit *cyp19a1a* to induce Sertoli cell development and male development. In MSM− individuals, lowly expressed Gsdf-A and Gsdf-B cannot inhibit *cyp19a1a*, leading to estrogen production, granulosa cell development, and female development. *In vitro*, Gsdf-A and Gsdf-B interact with Ncoa5 and blocks Ncoa5 interaction with Rora, resulting in the reduction of Rora/Ncoa5-induced activation on *cyp19a1a*. The expressions of *gsdf* and *cyp19a1a* are mainly in the somatic cells around germ cells.

As we known, Foxl2/Sf1-*cyp19a1a* pathway is an important pathway of female sex differentiation and knockout of *foxl2* results in female to male sex reversal in fish [25,66]. We suppose that the pathway of Rora/ Ncoa5 induced *cyp19a1a* activation is not independent of Foxl2/Sf1-*cyp19a1a* pathway. For instance, the *cyp19a1a* expression change induced by Foxl2/Sf1 can affect estradiol synthesis, which may lead to expression changes of *er* (*estrogen receptor*) and *rora* in the presence of Ncoa5 and then affect *cyp19a1a* transcription via Rora/Ncoa5-*cyp19a1a* pathway [61,62]. Thus, these two pathways of *cyp19a1a* regulation may interact with each other.

The sexual phenotype is the result of antagonism between the female and male pathways, with multiple feedback loops that are influenced by genotypic and/or environmental factors [37]. In hexaploid gibel carp, we have found that *gsdf* and *cyp19a1a* play antagonistic roles in sex differentiation. Gsdf represses *cyp19a1a* by blocking Ncoa5's availability for activation of *cyp19a1a* transcription. Meanwhile, *cyp19a1a* also has the ability to downregulate *gsdf* by suppressing *gsdf*'s transcription factor *dmrt1* [59]. Antagonistic actions of *dmrt1* and *foxl2* have been found in many other vertebrates [67–70] and these two genes also display conserved expression patterns during sex differentiation in gibel carp (Fig 5) [25,71]. In addition, Foxl2 also has been demonstrated to be a positive transcriptional factor of *cyp19a1a* as previously reported [60], indicating that Dmrt1 and Foxl2 may also play conserved antagonistic roles in gibel carp. As shown in previous studies, on one hand upregulation of *cyp19a1a* would lead to *dmrt1* inhibition while on the other it would upregulate *foxl2*.

In this study, we have identified two duplicated *gsdf* homeologous genes, *gsdf-A* and *gsdf-B*, and each homeolog has three alleles in the gynogenetic hexaploid gibel carp. The transcription of *gsdf-A* and *gsdf-B* is mainly activated by *dmrt1-A* and *dmrt1-B*, respectively. Moreover, loss-of-function experiments reveal the cooperative ability of two *gsdf* homeologs to regulate male differentiation by interacting *cyp19a1a* transcription. And the interactive mechanism analyses demonstrate that Gsdf interacts with Ncoa5 to suppress *cyp19a1a* transcription *in vitro*. This study provides a typical case of cooperative mechanism of duplicated genes in polyploids and also sheds light on the conserved evolution of sex differentiation.

## Materials and methods

### Ethics statement

Animal experiments and treatments were performed according to the Guidelines for Animal Care and Use Committee of Institute of Hydrobiology, Chinese Academy of Sciences (IHB, CAS, Protocol No. 2016–018).

### Fishes and cells

Experimental fish species including hexaploid gibel carp (*C. gibelio*) and red common carp (*C. carpio*) were provided and raised by the National Aquatic Biological Resource Center (NABRC), Institute of Hydrobiology, Chinese Academy of Sciences, Wuhan, China. Fish cell line *Carassius auratus* L. blastulae embryonic (CAB) cells and epithelioma papulosum cyprini (EPC) cells were maintained at 28˚C in 5% $CO_2$ in medium 199 (Invitrogen) supplemented with 10% fetal bovine serum (FBS) (Invitrogen).

### Cloning and sequence analysis

The divergent *gsdf* homeologs including *gsdf-A* and *gsdf-B* were identified according to the assembled genome of hexaploid gibel carp (*C. gibelio*) (Genbank accession numbers: PRJNA546443). Full-length cDNAs of *gsdf-A* and *gsdf-B* were obtained by 5' and 3' rapid

amplification of cDNA ends (RACE) (SMARTer RACE 5'/3' Kit, Clontech) using testicular cDNA library. Specific primers for RACE amplification (S2 Table) were designed according to the genome sequences of *gsdf-A* and *gsdf-B*. Multiple alignments of *gsdf* genomic and cDNA sequences was performed by DNAMAN 8.0 software.

The deduced amino acid sequences were predicted by DNAMAN 8.0 software. Multiple alignments of deduced amino acid sequences was performed by ClustalX program and exhibited by Bioedit program. Phylogenetic construction was adjusted by bootstrap analysis (1000 replicates) using the neighbor-joining method (NJ) in MEGA version 7.0 [72]. All the amino acid sequences of Gsdf in other fishes used in this analysis were obtained from Genebank. The accession number are as follows: *C. auratus* Gsdf-A, XP_026110442.1; *C. auratus* Gsdf-B, XP_026111149.1; *D. rerio* Gsdf, XP_017208308.1; *I. punctatus* Gsdf, XP_017347335.1; *O. latipes* Gsdf, BAJ05045.1 and *O. niloticus* Gsdf, BAJ78985.1. Syntenic analyses were conducted by comparing the chromosomic regions around *gsdf* genes in different fish species, including *C. gibelio* chromosomes (*Cg*A21 and *Cg*B21), *C. auratus* chromosomes (*Ca*A21 and *Ca*B21), *D. rerio* chromosome 21, *I. punctatus* chromosome 18, *O. latipes* chromosome 12, and *O. niloticus* chromosome 7. These information was obtained from GenBank (http://www.ncbi.nlm.nih.gov/, last accessed August 12, 2021).

## RNA extraction and qPCR

Adult organs, including heart, liver, hypothalamus, pituitary, kidney, spleen, ovary and testis, were isolated from three mature WT females (MSM-) and three mature WT males (MSM+), respectively. Male individuals were obtained from the offspring of a WT female mating with a WT male and all the female offspring were excluded by PCR detection using the male-specific marker. WT male gonads at different developmental stages were carefully dissected, and a total of 25, 25, 20, 10, 5, 5, 5, 5, 5, 5, 5, and 5 gonads were pooled for RNA extraction at the developmental stages of 17, 21, 30, 45, 60, 90, 120, 150, 210, 250, 300, and 360 dph, respectively. Besides, a total of 25, 5, 5 gonads from *gsdf* mutants and corresponding WT individuals were pooled for RNA extraction at the stages of 25, 55, 100 dph, respectively.

Total RNA isolation was performed using SV Total RNA isolation System (Promega), and the isolated RNAs were reverse-transcribed by the PrimeScript RT Reagent Kit (Takara). qPCR was performed on S1000 Thermal Cycler (BioRad), using iQSYBR Green Supermix (BioRad) as described previously [73]. *β-actin* was used as internal reference. All samples were analyzed in triplicates, and relative expression level of target gene was calculated with $2^{-\Delta\Delta CT}$ method. The highest expression level in each qPCR analysis was used as control and defined as 1 separately. Data were displayed as mean ± standard deviation. Significant differences were calculated by one-way ANOVA followed by Tukey test.

## Polyclonal anti-Gsdf antibody preparation and western blot

The *gsdf-A* cDNA sequence coding for 170 amino acids (S1 Fig) was cloned into Frd-GST vector (Friendbio Science and Technology) and the prokaryotic fusion protein was used as antigen to immunize a rabbit. Anti-Gsdf polyclonal antibody was produced by Friendbio Science and Technology Company Limited (Wuhan). Sample protein was extracted from cells using RIPA Lysis Buffer (Beyotime). Western blot detection was performed according to the previous reports and β-actin was used as internal control [74]. The images were obtained by Image-Quant LAS 4000mini (GE).

## Histological analysis and immunofluorescence

The gonads of gibel carp were fixed with 4% paraformaldehyde in PBS at 4˚C over night. After washing with PBS, the samples were immersed in 30% saccharose-PBS buffer for 5 h at 4˚C,

embedded in paraffin, and then were cut into 4μm sections. Hematoxylin-eosin staining was performed as described previously [50]. Immunofluorescence co-localization of Gsdf and Vasa was performed as described previously [71]. The images were obtained by upright fluorescence microscope Axio Imager M2 (Carl Zeiss).

## Fluorescence *in situ* hybridization (FISH)

Probes for *ncoa5* and *dmrt1* antisense/sense digoxigenin-labeled RNA strands were transcribed *in vitro* using the DIG RNA labeling kit (Roche). Probes for *roar*, *cyp19a1a*, and *sf-1* antisense/sense fluorescein-labeled RNA strands were transcribed *in vitro* using the Fluorescein RNA labeling kit (Roche). Specific primers with a T7 RNA polymerase promoter were designed to amplify complementary DNA (cDNA) fragment of each gene (S2 Table). Each probe was used at a final concentration of 0.5ng/μL. For more sensitive fluorescence in situ hybridization detection, the tyramide signal amplification TSA Plus Cyanine 3/Cyanine 5 System (PerkinElmer Life Science) was used according to the manufacturer's instructions. Digoxigenin-labeled RNA was stained with cy5, fluorescein -labeled RNA was stained with cy3. FISH analyses using sense RNA strands were shown in S12 Fig.

## Evolutionary analysis of *gsdf* or *dmrt1* homeologs

To investigate the potential role of selection on the evolution of *gsdf* or *dmrt1* homeologs, the *gsdf* or *dmrt1* gene dataset was assessed by branch model tests [75]. Alternative branch models, which allow foreground and background lineages to evolve differently (with different dN/dS), were compared to null models that assume the same ratio for all branches. Tree with divergence time was taken from MEGA version 7.0 [76]. The alternative models were evaluated for statistically significance ($P < 0.05$) by likelihood ratio tests (LRTs), with the null model using a $\chi2$ distribution [77].

## Generation of *gsdf* mutants by CRISPR/Cas9

Mutant line of *gsdf-A* and *gsdf-B* were generated by CRISPR/Cas9 as described previously [25] and the process was shown in S5 Fig. The sgRNA target sites of *gsdf-A* and *gsdf-B* were designed on the first exon and the second exon, respectively (Fig 1A and 1B). gRNAs were transcribed with the TranscriptAid T7 High-Yield Transcription Kit (Thermo Fisher Scientific). The gRNA and Cas9 protein (Invitrogen) were co-injected into one-cell-stage embryos at a concentration of 200 ng/μL and 100 ng/μL, respectively.

## DNA extraction and PCR detection of MSM

A small piece of fin was used to extract genomic DNA for each sampled fish, using DNA extraction kit (Promega) according to the manufacturer's instructions. The MSM was detected by PCR using the primer pair *Cg*-MSM-F and *Cg*-MSM-R [32]. PCR analysis was performed as previously described [35].

## Aromatase inhibitor treatment

Individuals from a gynogenetic family (WT, MSM−) and individuals from a *gsdf-A*/*gsdf-B* double mutant family (*gsdf-A*$^{\Delta4,+9/\Delta4/\Delta7}$ + *gsdf-B* $^{\text{chimeric mutations}}$, MSM+) were divided into two groups respectively, including a control group and a treatment group. The treated fish fry were fed with fairy shrimp that had been placed in 95% ethanol containing Letrozole (MCE) at a final concentration of 150 μg/L for 0.5 hr, whereas the control fish were fed with fairy shrimp that had been socked with 95% ethanol only. The treatment lasted for 40 days from 15 to 55

dph, and then all groups were fed with normal diet and maintained in outdoor tanks as described previously [32,78].

## Plasmid construction

According to genome database of gibel carp, the upstream sequences of *gsdf*–A (from -2080 to +50), *gsdf-B* (from -2150 to +50), and *cyp19a1a-B* (from -2066 to +50) were amplified from the genomic DNA and cloned into pGL3-Basic luciferase reporter vector (Promega). The open reading frame (ORF) of *gsdf-A*, *gsdf-B*, *dmrt1-A*, *dmrt1-B*, *sf1-A*, *sf1-B*, *ncoa5-B*, *roraα-B*, *foxl2b-A*, and *foxl2b-B* were amplified from testicular cDNAs of mature gibel carp by PCR and cloned into pcDNA3.1(+) vector, separately. *Roraα-B* and *ncoa5-B* ORF with N-terminal hem-agglutinin (HA)-tag were cloned into pCGN-ham vector, while *gsdf-A*, *gsdf-B*, and *ncoa5-B* ORF with N-terminal Flag-tag were cloned into pCMV-Tag 2 vector (Agilent Technologies). G*sdf-A*, *gsdf-B*, *ncoa5-B*, *roraα-B*, *foxl2b-A*, and *sting* ORF with C-terminal Myc-tag were cloned into pcDNA3.1/*myc*-His(-) A Vector (Invitrogen). The full-length coding sequences of 27 genes confirmed by yeast two-hybrid assay were cloned into pcDNA3.1(+) vector, separately. All constructs were confirmed by sequence analysis.

## Transient transfection

CAB cells were cultured in 6-well plates of phenol red-free M199 media (Gibco) supplemented with 10% charcoal dextran-treated serum (BI) until the cultures became approximately 75% confluent. Confluent cells were transfected using FuGENE HD Transfection Reagent (Promega) with 2 μg expression vectors. To analyze how Gsdf inhibits *cyp19a1a* transcription, we added 17β-estradiol (E2; Sigma-Aldrich) to the cells at a final concentration of 10 nM at 4 h post-transfection to elevate *cyp19a1a* transcription. The cells were harvested for RNA extraction and subsequent qPCR analysis. All the experiments were performed in triplicates.

## Luciferase activity assays

CAB cells (or EPC cells) were seeded in 24-well plates of phenol red M199 medium and co-transfected with various plasmid constructs at a ratio of 10:10:1 (250 ng luciferase reporter gene plasmid: 250 ng expression plasmid: 25 ng Renilla luciferase plasmid pRL-TK) using FuGENE HD Transfection Reagent (Promega). Then, transfected cells were harvested at 24 h post-transfection and measured by the Dual-Luciferase Reporter Assay System (Promega). Luciferase activities were measured by a Junior LB9509 luminometer (Berthold, Pforzheim, Germany) and normalized to the amounts of Renilla luciferase activities. All experiments were performed at least 3 times and the significant differences were calculated by SPSS soft-ware (SPSS Inc.).

## Yeast two-hybrid assay using DUAL membrane system

RNAs from female (MSM−) and male (MSM+) gonads at 15, 18, 22, 26, 30, 35, 40, 47, 58, 70, 110, 200, and 360 dph were pooled and reverse-transcribed. The purified double strand cDNAs were cloned into pDONR222 (Invitrogen) by BP Clonase II enzyme mix (Invitrogen) and the library titre was $1.12 \times 10^7$ cfu (colony-forming units) (S13A, S13B and S13E Fig). Then, these cDNAs were transferred from plasmid pDONR222 to pPR3-N-DEST (Dualsystems Biotech) by BP Clonase II enzyme mix (Invitrogen) and the library titre was $3.40 \times 10^7$ cfu (colony-forming units) (S13A, S13C and S13F Fig). The mature peptide sequence of *gsdf-A* (from 95 to 186 amino acids) was cloned into pBT3-SUC vector (Dualsystems Biotech) as bait

(pBT3-SUC -Gsdf-A) and yeast library screening was performed according to protocol of the DUAL membrane starter kits User Manual (Dualsystems Biotech).

## Yeast two-hybrid assay using GAL4 system

cDNAs were transferred from plasmid pDONR222 to pGADT7-DEST via homologous recombination by LR Clonase II Mix (Invitrogen) (S13A, S13D and S13G Fig). The extracted pGADT7-DEST-cDNA plasmids were transfected into yeast competent cell Y187, the library titre was $1.60 \times 10^7$ cfu. The library titer was calculated as described previously. Insert size was identify by PCR using primer pair pGADT7-F (T7) and pGADT7-R (ADR) (S2 Table). Subsequently, the well-distributed yeast cells were cultured on 100 plates (150 mm) with dropout medium (SD/-Leu) at 30°C for 5 days. All the appeared colonies were collected into freezing medium (YPDA medium with 25% glycerol) and stored at −80°C for yeast two-hybrid screening. The above construction of cDNA libraries were performed by OE BioTech (Shanghai, China).

The coding sequence of *gsdf-A* was cloned into pGBKT7 vector (Clontech) as bait (pGBTKT7-Gsdf-A), and yeast library screening was performed according to protocol of the Matchmaker Gold Yeast Two-Hybrid System (Clontech). All positive colonies were collected from quadruple dropout medium (QDO supplemented with X-alpha-Gal and Aureobasidin A, lack of Ade, His, Leu, and Trp) separately and used for plasmid extraction. Each plasmid was transformed into TOP10 chemically competent cell for subsequent sequencing. To exclude false positive results, the full-length coding sequences of all candidate genes were constructed into pGADT7 (Clontech) and then co-transformed to Y2H competent cell with pGBKT7-Gsdf-A on QDO/X-alpha-Gal/AbA plates and DDO (lack of Leu and Trp) plates, respectively.

## RNA interference

EPC cells were cultured in 12-well plates overnight, and then transfected with 50 nM small interfering RNAs (siRNA) of *ncoa5* (*ncoa5-A* and *ncoa5-B*) and the negative control (si-Nc) by using FishTrans (MeiSenTe Biotechnology). 17β-estradiol was added to the cells at a final concentration of 10 nM at 4 h post-transfection to elevate *cyp19a1a* transcription. siRNA of *ncoa5* (*ncoa5-A* and *ncoa5-B*) and si-Nc were synthesized by Sangon Biotech (Shanghai). The following sequences were targeted for *ncoa5* (*ncoa5-A* and *ncoa5-B*): si-*ncoa5*: CCGUCAUAGUC GUCAACAATT.

## Co-immunoprecipitation assay

CAB cells (or EPC cells) were seeded in 10 cm² dishes overnight and then transfected with a total of 10 μg of various plasmid combinations. The transfected cells were washed twice with 10 mL ice-cold PBS and then lysed by radioimmunoprecipitation (RIPA) lysis buffer with protease inhibitor cocktail (Sigma-Aldrich). After removing cellular debris, the supernatant was transferred to a 1.5 mL clean tube and incubated with 25 μL anti-Flag Affinity gel (Sigma-Aldrich) or anti-HA Magnetic Beads (Thermo Fisher) overnight at 4°C with constant rotating incubation. Immunoprecipitated proteins were collected by Magnetic Stand (Promega), washed five times with lysis buffer, and resuspended in 100 μL SDS-PAGE protein loading buffer (Beyotime). The immunoprecipitates and whole cell lysates (WCLs) were separated by 10–12% SDS-PAGE and then transferred to polyvinylidene fluoride membranes (Millipore) for subsequent western blot analysis. Antibodies were diluted as follows: anti-β-actin (Cell Signaling Technology) at 1:3,000, anti-Flag/HA antibody (Cell Signaling Technology) at 1:3,000, anti-Myc antibody (Abcam) at 1:2,000, and HRP-conjugated anti-rabbit IgG (Thermo

Scientific) at 1:5,000. Images were captured by ImageQuant LAS 4000mini (GE). Results were representative of three independent experiments.

### Chromatin immunoprecipitation (ChIP)

CAB cells were cultured in 15 cm$^2$ plate overnight and then transfected with 20 μg expression vector pCS2+-Rora-Myc. The transfected cells were used for ChIP analysis by ChIP-IT Express Chromatin Immunoprecipitation Kits (Active Motif). The protein and chromatin complexes were immunoprecipitated by 3 μg Anti-Myc antibody (Abcam) or anti-IgG antibody (Dia-An Biotec), respectively. The purified DNA was used for PCR analysis after Immunoprecipitation purification. The PCR products were electrophoresed in 2% agarose gels. The primer pair Chip-*rora*-F and Chip-*rora*-R (S2 Table) was used to amplify specific region spanning the potential binding site for Rora.

### Supporting information

**S1 Fig. Sequence alignments and phylogenetic construction. (A)** Coding sequence alignment of three *gsdf-A* alleles and three *gsdf-B* alleles. Sequence lengths and identities are exhibited at the end of sequences. **(B)** Multiple amino acid sequence alignment of Gsdf proteins from different fish species. Star marks the conserved cysteine. TGF-β superfamily domain is highlighted by black box. Sequences highlighted by red box were used as antigen for anti-Gsdf antibody preparation. **(C)** Phylogenetic tree of Gsdf proteins from different fish species.
(TIF)

**S2 Fig. Specificity of polyclonal anti-Gsdf antibody confirmed by western blot.**
(TIF)

**S3 Fig. Sequence preference of Dmrt1 recognition motif and putative Dmrt1-binding sites. (A)** Sequence preference of Dmrt1 recognition motif (MA1603.1) from JASPAR database. **(B)** Information about putative Dmrt1-binding sites of *gsdf* promoter.
(TIF)

**S4 Fig. Sequence preference of Sf1 recognition motif and putative Sf1-binding sites. (A)** Sequence preference of Sf1 recognition motif (MA1540.1) from JASPAR database. **(B)** Information about putative Sf1-binding sites of *gsdf* promoter.
(TIF)

**S5 Fig. Establishment of *gsdf* mutant families. (A-B)** Construction of different *gsdf-A* mutant lines **(A)**, *gsdf-B* mutant lines **(B)**, and *gsdf-A/gsdf-B* double mutant families **(A, B)**. Sex reversed individuals are marked in red color and sex reversal rate was shown at the bottom of each line or family. ♀, phenotypic female; ♂, phenotypic male.
(TIF)

**S6 Fig. Aromatase inhibitor treatment on gynogenetic offspring. (A)** Gonadal histology of gynogenetic offspring without letrozole treatment at 100 dph. **(B)** Gonadal histology of gynogenetic offspring with letrozole treatment at 100 dph. Bar: 20 μm. I, primary oocyte; II, growth stage oocyte; SG: spermatogonium; PSP: primary spermatocyte; SSP: secondary spermatocyte; SC: somatic cell. MSM−, without MSM; WT, wild type.
(TIF)

**S7 Fig. FISH analysis of the *cyp19a1a* (Red) mRNA and immunofluorescence analysis of the Gsdf protein (Green) in mature testis.** Arrowhead indicates the somatic cells with

expression of *cyp19a1a* and Gsdf. Scale bars: 25 μm.
(TIF)

**S8 Fig. Yeast two-hybrid assay using DUAL membrane system.** **(A)** Plate counting of 1,000-fold diluted yeast cells co-transformed with pBT3-SUC-Gsdf-A mature peptide and pPR3-N-library on DDO plate. The number of clones is displayed at the right bottom. **(B)** Undiluted yeast cells co-transformed with pBT3-SUC-Gsdf-A mature peptide and pPR3-N-library on QDO plate. No positive cells.
(TIF)

**S9 Fig. Transcription of sex differentiation genes in response to overexpression of Gsdf-A's potential interaction partners.** qPCR analysis of *cyp19a1a*, *foxl2b*, and *dmrt1* expression in the CAB cells transfected with different plasmids. Candidate genes represented by different numbers are given in Fig 6B. Different letters represent statistical differences (*$P<0.05$, **$P<0.01$, ***$P<0.001$). Ncoa5 activated *cyp19a1a* transcription but could not change expression levels of *foxl2b* and *dmrt1*.
(TIF)

**S10 Fig. Identification of *ncoa5* and *rora* homeologs in hexaploid gibel carp.** **(A-B)** Chromosomal localization of *ncoa5* **(A)** and *rora* **(B)**. Chromosome numbers are displayed at the left side. Conserved gene blocks are represented in matching colors. Transcription orientations are indicated by arrows. **(C-D)** Deduced amino acid sequence alignment of Ncoa5 **(C)** and Rora **(D)**. The identities relative to human orthologs are exhibited at the end of each sequence. Ncoa5-B and Roraα-B were selected for subsequent *in vitro* analyses.
(TIF)

**S11 Fig. Sequence preference of Rora recognition motif and putative Rora-binding sites.** **(A)** Sequence preference of Rora recognition motif (MA0071.1) from JASPAR database. **(B)** Information about putative Rora-binding sites of *cyp19a1a* promoter.
(TIF)

**S12 Fig. FISH analyses using sense RNA strands.** Sections of mature testis were subjected for FISH with sense riboprobes of *dmrt1* (Red), *ncoa5* (Red), *cyp19a1a* (Pink), *roar* (Pink), and *sf-1* (Pink), and analyzed by fluorescence microscopy. Scale bars: 25 μm.
(TIF)

**S13 Fig. Evaluation of cDNA libraries.** **(A)** Summary of three cDNA libraries using different plasmids including pDNOR222, pPR3-N-DEST, and pGADT7-DEST. **(B-D)** Plate counting of 200-fold diluted *E. coli* cells from the libraries of pDNOR222 **(B)**, pPR3-N-DEST **(C),** and pGADT7-DEST **(D)**. **(E-G)** Agarose gel electrophoresis of PCR products from randomly selected 24 colonies from the library of pDNOR222 **(E)**, pPR3-N-DEST **(F),** and pGADT7-DEST **(G)**. Marker is DL2000 DNA marker.
(TIF)

**S1 Table. Comparison of coding sequences and protein sequences between *gsdf-A* and *gsdf-B* alleles.**
(DOCX)

**S2 Table. Primers used in this study.**
(DOCX)

## Acknowledgments

We thank Fang Zhou and Yan Wang (the Center for Instrumental Analysis and Metrology, Institute of Hydrobiology, Chinese Academy of Science) for providing confocal services.

## Author Contributions

**Conceptualization:** Xi-Yin Li, Jian-Fang Gui.

**Formal analysis:** Ming-Tao Wang, Zhi Li, Miao Ding, Tian-Zi Yao, Sheng Yang, Xiao-Juan Zhang, Chun Miao, Wen-Xuan Du, Qian Shi, Shun Li, Jie Mei, Yang Wang, Zhong-Wei Wang, Li Zhou.

**Funding acquisition:** Li Zhou, Xi-Yin Li, Jian-Fang Gui.

**Investigation:** Ming-Tao Wang, Zhi Li, Miao Ding, Tian-Zi Yao, Sheng Yang, Xiao-Juan Zhang, Chun Miao, Wen-Xuan Du, Qian Shi, Shun Li, Jie Mei, Yang Wang, Zhong-Wei Wang, Li Zhou.

**Methodology:** Ming-Tao Wang.

**Resources:** Ming-Tao Wang, Xi-Yin Li.

**Writing – original draft:** Xi-Yin Li.

**Writing – review & editing:** Xi-Yin Li, Jian-Fang Gui.

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
