## [Decision Letter · Decision Letter 0]

13 Apr 2022

Dear Dr Gui,

Thank you very much for submitting your Research Article entitled 'Two duplicated gsdf homeologs cooperatively regulate male differentiation by inhibiting cyp19a1a transcription in a hexaploid fish' to PLOS Genetics.

The manuscript was fully evaluated at the editorial level and by independent peer reviewers. The reviewers appreciated the attention to an important problem, but raised some substantial concerns about the current manuscript.Based on the reviews, we will not be able to accept this version of the manuscript, but we would be willing to review a much-revised version. We cannot, of course, promise publication at that time.

Should you decide to revise the manuscript for further consideration here, your revisions should address the specific points made by each reviewer, in particular those concerning the subgenome-specific regulation of gsdf.  We will also require a detailed list of your responses to the review comments and a description of the changes you have made in the manuscript.

If you decide to revise the manuscript for further consideration at PLOS Genetics, please aim to resubmit within the next 60 days, unless it will take extra time to address the concerns of the reviewers, in which case we would appreciate an expected resubmission date by email to plosgenetics@plos.org.

[LINK]

We are sorry that we cannot be more positive about your manuscript at this stage. Please do not hesitate to contact us if you have any concerns or questions.

Yours sincerely,

Manfred Schartl

Guest Editor

PLOS Genetics

Gregory Barsh

Editor-in-Chief

PLOS Genetics

Reviewer's Responses to Questions

**Comments to the Authors:**

Reviewer #1: Review about Wang et al.; PLoS Genetics

Two duplicated gsdf homeologs cooperatively regulate male differentiation by inhibiting cyp19a1a transcription in a hexaploid fish.

In their study, Wang et al. have identified two gsdf homeologous genes in the hexaploid gibel carp. They further propose to better characterize their regulation and function regarding to gonad development.

First, having in mind to decipher how duplicated genes co-regulate a biological process (gonadal development for instance) in a polyploid species, they show that the transcription of gsdfA and gsdfB is mainly activated by dmrt1a and dmrt1b respectively, suggesting asymmetrical evolution to explain the expression divergence in the context of the two sub-genomes. Then, functional characterization is carried on through generation of gsdf mutants, leading to partial or total sex reversion. The cooperative ability of the two gsdf homeologs to regulate male differentiation is then shown. Using cell culture system, gsdf interactions with Ncoa5 and subsequent transcriptional modulation of Cyp19a1a is further demonstrated.

All in one, although the whole story presented in this paper draft is interesting in terms of (i) evolutionary biology (how duplicated genes co-regulate a biological process, and how asymmetrical evolution rule expression divergence), and (ii) mechanistic/physiology (how gsdfs exert their function(s)), I think some major issues need to be addressed before this draft is ready for publication in PLoS Genetics.

Major Issues:

-Characterization of gsdf homeologs and alleles/dynamic transcription of gsdfA/Bare mainly activated by dmrt1A/B respectively.

Co-evolution of the two subgenomes (regarding to gsdfs regulation by Dmrt1) is indeed an interesting aspect of that paper.

-The analysis might benefit from a dN/dS analysis (gsdf and dmrt1) to check whether one or the other homeologs are under selection. This might tell us more about the co-evolution. It should be then discussed if necessary.

-In silico analysis: authors need to tell us more about the predicted dmrt1/sf1-binding sites (which matrix was employed, sequence, homology, divergence to the canonical sequence…)

-At the end, we need to know if this differential regulation is due to the evolution (divergence) of the Dmrt1s only, or of the targets only, or co-evolution of both.

-Cytoplasmic localization of the Gsdf proteins is OK for me, although we know it is a secreted protein that will diffuse from one cell to others, and bind at cell membranes. It might accumulate in some cells before being excreted; this should be told and discussed.

-Disruption of gsdfA/B results in cyp19a1a upregulation and aromatase inhibitor treatment rescues the male to female sex reversal.

That part is OK for me.

-Figure 3A/3B is very complex and I am not sure it helps much here. I would move it to supplemental data.

-Figure 4E: it is difficult to compare that figure 4E with what has been shown in figure 2B: in 2B, basal relative expression of Gsdfs is nearly not detectable (at 25 dph), while that basal expression is quite high in 4E (same stage). Reference to (E) is actually not well written in the figure legend.

So, please unify the relative expressions (2B/4E) or explain that discrepancy.

The rest is OK for me, although at one point you will need RNA in situs to show that Gsdf, Ncoa5 and rora are indeed co-expressed to validate the physiological relevance of the inferred interactions.

-Identification of GSDF-NCOa5 interaction via yeast two hybrid assay and co-immunoprecipitation.

-Well, although I could be at first convinced by the yeast two hybrid assay, I am nevertheless very concerned about the fact that the full open reading frame of Gsdf was used as a bait. Indeed, if I am right the whole ORF usually encodes for the pro-mature peptide. Then to be active the pro-domain has to be cleaved to release the mature gsdf active peptide. In these conditions, using the un-cleaved Gsdf, I am not certain that the found interactions are relevant. This is an issue.

-The same question now stands for the antibody that was prepared from (most of) the whole sequence; and that might also recognize the pro-domain instead of the mature gsdf peptide.

-NCOa5 participates in cyp19a1a regulation via interaction with Rora.

-Like for Sf1 and Dmrt1 predicted binding sites, potential Rora binding sites should be described with more details: statistics concerning the homology to the matrix and so on.

The rest of that part seems OK for me.

-Ncoa5 participates in cyp19a1a regulation via interaction with Rora.

-Regulations and interactions were established in cells lines after transfection experiments. While the experiments performed for that part are quite OK, we should not forget the physiology behind, and how to interpret these results “physiologically”.

Indeed, while Figure 8 compile the whole story, it is difficult for me to understand how a TGF-beta ligand, that is an excreted molecule that usually binds at the cell surface, would here have an activity inside of the nucleus together with Ncoa5 and Rora. To my knowledge this is impossible.

Then to make the necessary connexion between the results in cell lines together with what the physiology would allow, it is to my opinion necessary to show co-localization of the Gsdf-Ncoa5-Rora complexes inside of the nucleus.

Reviewer #2: Gsdf is a member of transforming growth factor β (TGF-β) superfamily which has been implicated in male sex differentiation of multiple teleost species. Despite of its critical role in testis differentiation, the specific molecular pathway of Gsdf-mediated sex differentiation remains elusive. In this MS, the authors identified two duplicated gsdf homeologs genes (gsdf-A and gsdf-B) in the gynogenetic hexaploid gibel carp, and found that the transcription of gsdf-A and gsdf-B is activated by dmrt1-A and dmrt1-B, respectively. By loss-of-function experiments, the authors demonstrated that two gsdf homeologs cooperatively regulate male differentiation by inhibiting cyp19a1a transcription. By in vitro analyses, the authors demonstrated that Gsdf-A and Gsdf-B interact with Ncoa5 to block Ncoa5 interaction with Rora, inhibiting Rora/Ncoa5-induced activation of cyp19a1a.

Generally, this study is interesting and provides a special mechanism of gsdf in regulating male sex determination and differentiation. However, the following issues should be addressed before the article is accepted.

Major concerns:

1) As we know, Gsdf is a secretory protein which belongs to the transforming growth factor β family, and it is supposed to function with its receptor. In this MS, the author demonstrated a direct interaction between Gsdf-A/B and Ncoa5. Is there a TGF- β type II receptor involved in this regulatory process by binding gsdf?

2) Usually, the male and female pathway genes display obvious sexually dimorphic expression. If both Ncoa5 and Rora are involved in sex differentiation, why there is no difference in the expression of ncoa5 and rora between females and males in the Figure 7A. Further, in Figure 7B, overexpression of gsdf led to downregulation of roar, why there was no difference in the expression of rora between females and males in the Figure 7A?

3) An important finding of this work is that in females Ncoa5/Rora induced activation of cyp19a1a independent of Foxl2/Sf1-cyp19a1a pathway. We all know that mutation of Foxl2 lead to female to male sex reversal. The specific function of Ncoa5 in fish and whether it is involved in fish sex differentiation has not been explored. For this reviewer, it is hard to believe Ncoa5/Rora-cyp19a1a and Foxl2/Sf1-cyp19a1a pathway is independent in female sex differentiation.

4) In Figure 2D and 2E, the authors demonstrated that Dmrt1 and Sf1 activate gsdf expression through binding to their potential binding sites in the gsdf promoter, it is better to determine which site is important for gsdf activation. It is also better to demonstrate that Dmrt1 and Sf1 are co-expressed in gsdf expressing cells.

Miner concerns:

Line 172：“Expression plasmids of Sf1-A/Sf1-B, Dmrt1-A, Dmrt1-B, and an empty expression plasmid were used as controls” should be “Expression plasmids of Sf1-A/Sf1-B, Dmrt1-A, Dmrt1-B were constructed, and an empty expression plasmid was used as controls”.

Line 555: “mediaand” should be “medium”.

Line 556: “luc” should be “luciferase reporter gene”.

Line 633: A repeated copy of “under the accession number of” should be deleted.

Line 904: “exam” should be “examine”.

Line 908: “indicates” should be “indicate”.

Line 921: “transcriptions” should be “transcripts” or “mRNA expression”, and the same for line 934.

Line 958: The letter “e” in the Figure legend should be “(E)”.

Line 980: “activates the promoter activity of gsdf-A/gsdf-B”, the promoter described here should be the “cyp19a1a promoter”.

Line 1071：“noca5” should be “ncoa5”.

Reviewer #3: This is an interesting and well-conducted study showing the role of a member of the transforming growth factor beta (TGF-ß) signaling pathway, gonadal somatic cell-derived factor (gsdf) to be involved in male differentiation in the hexaploid gibel carp (Carassius gibelio). This species was used as a system to investigate the mechanisms that duplicate genes use in a developmental process such as sex differentiation. Authors demonstrate the presence of two homeologues of gsdf, termed gsdf-A and gsdf-B that cooperatively regulate male sex differentiation by interacting with the nuclear co-activator factor 5 (Ncoa5) to suppress gonadal aromatase (cyp19a1a) gene expression. In lower vertebrates, it is well established that aromatase activity is necessary for estrogen production and female development. The study is very complete and includes the molecular characterization of gsdf-A and gsdf-B genes, expression during gonad development, transcriptional regulation, and functional analysis by wisely combining induced gynogenesis using males with or without a male determinant contained in supernumerary microchromosomes and loss-of-function analysis by the production of single or double mutants by means of CRISPR technology. Authors convincingly show that disruption of either gsdf-A or gsdf-B results in partial sex reversal while disruption of both homeologs results in complete male-to-female sex reversal. Furthermore, male-to-female sex reversal in double mutants is due to stimulation of cyp19a1a, the effects of which can be inhibited by treatment with a non-steroidal aromatase inhibitor. Authors identify Ncoa5 as interaction partner of Gsdf and show that gsdf binding of Ncoa5 prevents the latter to interact with Ncoa to upregulate cyp19a1 transcription. The study nicely shows how both forms of gsdf cooperate towards male sex differentiation. I have some comments that may help to improve the ms.

L27 and Throughout the ms. The term “homeologous”, used in polyploidy plant research, may not be familiar with some readers, who may confuse it with “homologous”. Please explain that homologous genes resulting from allopolyploidy are commonly referred to as “homoeologs”. Here you use the term “homeologs”, which I am not sure how common it is. All this may create confusion. Please provide a clear definition of the term and its possible variants right from the start.

L. 91-93. “a variant of gynogenesis”. This needs to be better explained to readers. Please indicate ploidy level of each progenitor in the initial cross. Also explain whether there is a dose-dependent relationship between the microchromosomes and the percent of males.

L109. I am not sure if it would be more appropriate to speak about “sex determination” rather than to “sex differentiation”. Duplicates of gsdf has been implicated in sex determination. In fact, in lines 101-102 authors write “Members of the transforming growth factor-β (TGF-β) signaling pathway have been identified as being vastly involved in sex determination”.

L151. Gsdf-A expression during sex differentiation peaked at 250 dph while the expression of Gsdf-B peaked at 120 dph. However, gonadal morphological differentiation between males and females occurs around 40 dpf. How then it is possible that Gsdf-A and B are drivers of sex determination/differentiation?

L166-168. Consider rephrasing the sentence to clearly indicate that the identified upstream sequences of gsdf-A and gsdf-B were considered the potential promoter sequences; otherwise it seems that the binding sites per se are the promoter regions.

L191. Indicate how did you ensure that the female gibel carp was devoid of MSM.

L263. Delete “obviously”.

L276. Add “nonsteroidal” before “aromatase inhibitor”.

L277. Add “enzyme activity” after “Cyp19a1a”.

L279. Explain why there were approximately 15% of the fish in which male-to-female sex reversal could not be prevented.

L282. Upregulation of foxl2b in the testis of gsdf-A/gsdf-B double mutants (MSM+) subjected to letrozole treatment comes as a bit of surprise since usually foxl2a and Cyp19a1a expression goes hand in hand due to the regulatory positive feedback loop involving these two genes. Please clarify.

L317. How can you be sure that Ncoa5-A had no role in sex differentiation

L394. It comes as a bit of surprise that no further explanation of the different functions of gsdf-A and gsdf-B are discussed. This is relevant to the study but these functions are not properly explained. This is relevant in view of the overall aim of the paper. It should be stated whether the contribution of each homeologue is similar, if there is any indication that one has more relevance than the other in male differentiation.

L421. “Thus, dysfunction of gsdf triggers upregulation of cyp19a1a and subsequently leads to the inhibition of dmrt1 and activation of foxl2”. I would rephrase a bit this sentence because in this study the whole chain of events is not shown. So I suggest indicating that, as shown in previous studies, on one hand upregulation of cyp19a1a would lead to dmrt1 inhibition while on the other it would upregulate foxl2.

**Have all data underlying the figures and results presented in the manuscript been provided?**

Reviewer #1: Yes

Reviewer #2: Yes

Reviewer #3: Yes

PLOS authors have the option to publish the peer review history of their article (what does this mean?). If published, this will include your full peer review and any attached files.

Reviewer #1: No

Reviewer #2: **Yes: **Deshou Wang

Reviewer #3: No

---

## [Decision Letter · Decision Letter 1]

8 Jun 2022

Dear Dr Gui,

We are pleased to inform you that your manuscript entitled "Two duplicated gsdf homeologs cooperatively regulate male differentiation by inhibiting cyp19a1a transcription in a hexaploid fish" has been editorially accepted for publication in PLOS Genetics. Congratulations!

Before your submission can be formally accepted and sent to production you will need to correct the minor mistakes identified by reviewer #2 and complete our formatting changes, which you will receive in a follow up email. Please be aware that it may take several days for you to receive this email; during this time no action is required by you. Please note: the accept date on your published article will reflect the date of this provisional acceptance, but your manuscript will not be scheduled for publication until the required changes have been made.

Yours sincerely,

Manfred Schartl

Guest Editor

PLOS Genetics

Gregory Barsh

Editor-in-Chief

PLOS Genetics

Comments from the reviewers (if applicable):

Reviewer's Responses to Questions

**Comments to the Authors:**

Reviewer #1: Well, all my concerns have been addressed and discussed, although it was requiring quite a bit of additional work.

I do think this paper draft has been much improved, and that it is now ready for publication.

I have to add that it is a nice piece of work, and that is was real pleasure to review it.

Congratulation!

Reviewer #2: In this revised manuscript, the authors have carefully considered the reviewer’s comments and provided new evidences. Co-localization results of dmrt1/sf-1 with Gsdf along with mutation of Dmrt1 binding sites on gsdf promoter further supported the regulation of Dmrt1 on gsdf promoter in vitro. Additionally, new in situ expression data of rora/ncoa5 showed their co-expression with Gsdf and Cyp19a1a, respectively. These results strongly supported the regulation of Rora/Ncoa5 on cyp19a1a and validated the physical interaction between Gsdf and Ncoa5. Together the additional experiments and careful editing have significantly improved the quality of the manuscript. Overall, the authors have taken all the suggestions in consideration and have satisfactorily addressed all of my concerns. In one word, the authors have done a commendable job at revising the manuscript and have successfully improved it.

Attached below were a few minor mistakes:

Line 276: “are shown” should be “were shown”.

Line 611: The process concerning generation of gsdf mutants has been moved into revised Figure S5, it is not in Figure 3 now. Please revise “shown in Fig 3.” as “shown in S5 Fig.”

Line 613: “Fig 1A and 3B” here should be “Fig 1A and 1B”.

Line 644: “roarα-B” should be “roraα-B”.

Reviewer #3: The authors have successfully addresed all comments and clarified the doubts. This is a good paper suitable for publication.

**Have all data underlying the figures and results presented in the manuscript been provided?**

Reviewer #1: Yes

Reviewer #2: Yes

Reviewer #3: Yes

PLOS authors have the option to publish the peer review history of their article (what does this mean?). If published, this will include your full peer review and any attached files.

Reviewer #1: **Yes: **Amaury HERPIN

Reviewer #2: **Yes: **Deshou Wang

Reviewer #3: No

**Data Deposition**

http://datadryad.org/submit?journalID=pgenetics&manu=PGENETICS-D-22-00317R1

**Press Queries**

---

## [Editor Report · Acceptance letter]

25 Jun 2022

PGENETICS-D-22-00317R1 

Two duplicated *gsdf* homeologs cooperatively regulate male differentiation by inhibiting *cyp19a1a*transcription in a hexaploid fish 

Dear Dr Gui, 

We are pleased to inform you that your manuscript entitled "Two duplicated *gsdf* homeologs cooperatively regulate male differentiation by inhibiting *cyp19a1a*transcription in a hexaploid fish" has been formally accepted for publication in PLOS Genetics! Your manuscript is now with our production department and you will be notified of the publication date in due course.

With kind regards,

Zsofia Freund

PLOS Genetics

On behalf of:
